# On Weak Regret Analysis for Dueling Bandits

**El Mehdi Saad**[*]
KAUST
mehdi.saad@kaust.edu.sa

**Alexandra Carpentier**
Institut für Mathematik
Universität Potsdam
carpentier@uni-potsdam.de

**Tomáš Kocák**
Institut für Mathematik
Universität Potsdam
kocak@uni-potsdam.de

**Nicolas Verzelen**
INRAE, MISTEA, Univ. Montpellier
nicolas.verzelen@inrae.fr

## Abstract

We consider the problem of $K$-armed dueling bandits in the stochastic setting, under the sole assumption of the existence of a Condorcet winner. We study the objective of weak regret minimization, where the learner doesn't incur any loss if one of the selected arms is a Condorcet winner—unlike strong regret minimization, where the learner has to select the Condorcet winner twice to incur no loss. This study is particularly motivated by practical scenarios such as content recommendation and online advertising, where frequently only one optimal choice out of the two presented options is necessary to achieve user satisfaction or engagement. This necessitates the development of strategies with more exploration. While existing literature introduces strategies for weak regret with constant bounds (that do not depend on the time horizon), the optimality of these strategies remains an unresolved question. This problem turns out to be really challenging as the optimal regret should heavily depend on the full structure of the dueling problem at hand, and in particular on whether the Condorcet winner has a large minimal optimality gap with the other arms. Our contribution is threefold: first, when said optimality gap is not negligible compared to other properties of the gap matrix, we characterize the optimal budget as a function of $K$ and the optimality gap. Second, we propose a new strategy called WR-TINF that achieves this optimal regret and improves over the state-of-the-art both in $K$ and the optimality gap. When the optimality gap is negligible, we propose another algorithm that outperforms our first algorithm, highlighting the subtlety of this dueling bandit problem. Finally, we provide numerical simulations to assess our theoretical findings.

## 1 Introduction

We consider an instance of the problem of sequential and active learning from comparisons - namely dueling bandits. It can be modeled as a sequential game where, at each time, a learner presents to a user a pair of two items and collects feedback, which is a noisy indication of the user's preference between the two items. If neither of the presented items aligns with the user's top choice, the learner incurs a loss. Preference-based learning has gained importance recently as it reflects human decision-making processes, which often rely on relative rather than absolute evaluations. This approach is notably effective in systems that involve human interaction, where feedback is provided in a qualitative form [7]. In the dueling bandit setting, initially presented by [19], the compared items

---

[*]Work done while at CentraleSupélec.

38th Conference on Neural Information Processing Systems (NeurIPS 2024).

are called "arms" and there are $K$ of them. This setting is structured as an ongoing sequential game with time horizon $T$, where in each round $t \leq T$, the learner selects two arms (items) indexed by $i, j \in [K]$, and receives the result of a duel between these arms as feedback. The result of each duel is encoded by 1 if the first arm - in our case arm $i$ - beats the second one - in our case arm $j$ - and 0 otherwise. This result follows a Bernoulli distribution with unknown parameter $q_{i,j}$. Specifically, this paper focuses on the stochastic scenario where the parameters governing the duels are assumed to be constant throughout the game, albeit unknown to the learner. The literature on the stochastic dueling bandits is very rich and contains various settings that differ from our paper either in optimal arm characterization or the definition of rewards.

In contrast to the classic multi-armed bandit problem, defining the optimal arm in dueling bandits is not straightforward. This led to the introduction of various definitions of winners within the literature as detailed in the surveys [2, 16, 17]. Our study focuses on situations where there is an arm $k^* \in [K]$ that, on average, defeats all other arms: formally, $q_{k^*,j} > 1/2$ for any $j \in [K] \setminus \{k^*\}$. This arm is termed the Condorcet winner while we refer to it as optimal in the rest of the paper. In the context of dueling bandits, particularly the cumulative regret minimization problem, most of prior works either made the assumption of the existence of a Condorcet winner [24, 23, 10, 8, 4, 14, 15] or the stronger assumption of the exitence of a total order between arms [18, 20, 4].

Once the concept of the optimal arm is established, the next step is to define the objective. Rather than identifying the best arm, our goal is to minimize the cumulative loss. To this end, we must determine the loss incurred each round based on the two arms selected by the learner. The dueling bandit literature distinguishes between two primary types of losses: strong loss and weak loss, as described by [18]. With strong loss, the learner must select the Condorcet winner twice to avoid any loss (noting that the feedback in this case is equivalent to a fair coin flip). In contrast, weak loss requires only one of the selected arms to be the Condorcet winner. Formal definitions of weak and strong regrets are provided in Section 2. In many practical scenarios, such as recommendation systems and online advertising [6, 3], minimizing weak regret aligns more closely with the learner's objectives than strong regret minimization. For example, consider a situation where the learner operates as a service provider, presenting two options to a client who then chooses their preferred option. In this framework, the learner should incur a loss only if neither option matches the client's preference, encouraging exploration and maximizing information gain. While previous research in dueling bandits has primarily focused on minimizing strong regret, developing optimal strategies for minimizing weak regret remains an unresolved issue despite prior works [4, 12], as highlighted in the survey [2]. Further details on the technical distinctions between these two objectives will be discussed in the next sections.

In this paper, we focus on minimising the weak regret. We provide a lower bound for this problem in a specific regime where the Condorcet winner beats largely the other arms. We provide an algorithm that matches it. Nevertheless, it is not optimal in all regimes, and we highlight this by providing another algorithm that performs better in some interesting regimes.

## 2 Problem setting

We consider $K$ arms. Let $Q = (q_{i,j})_{1 \leq i,j \leq K} \in [0,1]^{K \times K}$ be the matrix of preference probabilities where the probability of arm $i$ beating arm $j$ in a duel corresponds to $q_{i,j}$. We assume that $q_{j,i} = 1 - q_{i,j}$ and $q_{i,i} = 1/2$ for all $i, j \in [K]$. Define $\Delta_{i,j} := q_{i,j} - \frac{1}{2}$. Notably, the sign of $\Delta_{i,j}$ indicates the relative preference between arms $i$ and $j$ (specifically, $i$ is preferred over $j$ if $\Delta_{i,j} > 0$). The quantity $\Delta_{i,j}$ characterizes the hardness of distinguishing which of the arms $(i, j)$ is preferred to the other. We denote $\mathbf{\Delta} := (\Delta_{i,j})$ the gap matrix. The only assumption made in this paper is regarding the existence of a Condorcet winner, which we denote $k^*$ for the remainder of this paper:

**Assumption 2.1.** *Existence of a Condorcet winner: There exists an arm $k^* \in [K]$ such that:*

$$\forall i \in [K] \setminus \{k^*\} : q_{k^*,i} > 1/2.$$

We consider that at each time $t = 1, 2, \ldots$, the learner chooses two arms $(I_t, J_t)$ based on past information and receives the output of a duel between the chosen arms. More formally, the output is a sample from a Bernoulli distribution with parameter $q_{I_t, J_t}$, independent of everything else after conditioning on $(I_t, J_t)$. We consider that after each round $t$ the learner incurs a loss given by:

$$\ell_t^{(w)} := \min\{\Delta_{k^*, I_t}, \Delta_{k^*, J_t}\},$$

which we term the weak instantaneous loss, following [18]. Another concept of instantaneous loss that is often considered in the literature is the strong instantaneous loss [18, 1, 24], where at round $t$ the learner incurs the loss $\ell_t^{(s)} := (\Delta_{k^*,I_t} + \Delta_{k^*,J_t})/2$. Note that when it comes to the weak instantaneous loss, in contrast to the strong instantaneous loss, the learner does not incur any loss if at least one of the two chosen arms is the Condorcet winner, $I_t = k^*$ or $J_t = k^*$, while for the strong instantaneous loss, both arms need to be the Condorcet winner in order for the learner to not incur a loss. We finally define the weak expected cumulative regret up to time $T$ by for the weak instantaneous loss by $R_T^{(w)} := \sum_{t=1}^T \mathbb{E}[\ell_t^{(w)}]$, which we term weak regret. Similarly, we define the strong regret as $R_T^{(s)} := \sum_{t=1}^T \mathbb{E}[\ell_t^{(s)}]$.

## 3 Literature review and our contributions

### 3.1 Related work

When it comes to the stochastic dueling bandit literature, much of the prior work has been devoted to the goal of minimizing the strong regret, under the assumption of a total order between the arms [19, 18] or only under the assumption of the existence of a Condorcet winner [24, 23, 10, 8, 12]. We detail nevertheless those results here as, since the strong regret upper bounds the weak regret, all algorithms and upper bounds that are available for the strong regret also hold for the weak regret. In [8], an instance-dependent lower bound for strong regret was established:

$$\liminf_{T \to \infty} \frac{\mathbb{E}\left[R_T^{(s)}\right]}{\log(T)} \geq \sum_{k \neq k^*} \min_{i \in \mathcal{O}_k} \frac{\Delta_{k^*,k} + \Delta_{k^*,i}}{2\Delta_{i,k}^2}, \tag{1}$$

where $\mathcal{O}_k = \{i \in [K] \mid q_{i,k} > 1/2\}$. This work also introduces an algorithm that asymptotically matches this lower bound as $T \to +\infty$. However, in finite-horizon, their regret bound has a quadratic dependence on the number of arms $K$. Deriving bounds that scale linearly with $K$ has been the subject of several works [23, 10], In particular, [14] devised a reduction to a standard (but adversarial) multi-armed bandits problem. They obtained guarantees on the strong regret which are of the order $\sum_{k \neq k^*} \log(T)/\Delta_{k^*,k}$. This regret bound turns out to match (1) in scenarios where the Condorcet winner is also the arm that is best for eliminating all other sub-optimal arms, namely where $\Delta_{k^*,k} = \max_i \Delta_{i,k}$. In more general cases, the last upper bound of [14] does not match the lower bound given in (1).

Weak regret itself was introduced in [18] to model in a more refined way some recommender systems applications. As mentioned, it is upper bounded by the strong regret so that all described algorithms and associated regret upper bounds would also hold for the weak regret. However, a distinction was made in [4] regarding the fundamental nature of these two problems. While the problem-dependent optimal order of the strong regret scales as $\log T$ (see above) - which is aligned with classical results in stochastic bandits - there exist some algorithms whose problem-dependent weak regret is upper bounded by a quantity that does not depend on $T$ - which is in sharp contrast with classical results on stochastic bandits. Specifically, [4] introduced an algorithm called WS-W, which, under the sole assumption of the existence of a Condorcet winner, achieves an upper bound on weak regret of the order $K^2/\min_{i \neq j} \Delta_{i,j}^2$. More recently, in [12], the Beat The Winner (BTW) algorithm was introduced. BTW adopts a round-based approach where the best arm so far keeps being challenged through batches of duels by candidate arms. Assuming only the presence of a Condorcet winner, this algorithm achieves an upper bound on weak regret of the order $K^2 + K/\min_{i \neq k^*} \Delta_{k^*,i}^4$. Finally, under the additional and arguably the strong assumption of the existence of a total order between arms, the upper bound can be proven to be of order $(K \log K)/\min_{i \neq j}|\Delta_{i,j}|^5$. In summary, the dependency on the optimal regret on both $K$ and on the matrix $\mathbf{\Delta}$ still remains largely unknown.

From a technical standpoint, developing optimal strategies in the weak regret framework underlies different challenges than the ones for the strong regret. This complexity arises because losses in the strong regret framework are linear in the problem parameters (the gaps matrix entries $(\Delta_{i,j})_{1 \leq i,j \leq K}$): $\ell_t^{(s)} = (\Delta_{k^*,I_t} + \Delta_{k^*,J_t})/2$, whereas in weak regret, the loss is determined as the minimum gap with the client's preference: $\ell_t^{(w)} = \min\{\Delta_{k^*,I_t}, \Delta_{k^*,J_t}\}$, which breaks linearity. As a result, classical reduction methods as the one used in [1, 14, 15] are not directly applicable for weak losses.

## 3.2 Main contributions

In this paper, we address the following fundamental question:

(i) What is the best possible weak regret one can achieve in terms of $K$ and the gaps $(\Delta_{k^*,i})_{i \neq k^*}$ to the Condorcet winner?

(ii) Beyond that, is it possible to improve the regret by leveraging over the *unknown* entries of the matrix $\mathbf{\Delta}$? As a simple toy example, assume that the gaps $\Delta_{k^*,k}$ are small and that, some gaps $(\Delta_{i,j})$ for $i, j \neq k^*$ are much higher. In that regime, it is perhaps more beneficial to explore the $\mathcal{O}(K^2)$ duels between all arms to better discard sub-optimal arms than simply to directly look for the Condorcet winner. This informal argument suggests the optimal guarantees depend in an intricate way on the number of arms $K$ and the gaps $(\Delta_{i,j})$ and that there is a complex trade-off between directly aiming for the Condorcet winner and further exploration for better elimination.

To address the first question, we provide a lower bound on the weak regret, which, to the best of our knowledge, is the first of its kind for this problem. We demonstrate that in certain cases, where in particular the gaps between the Condorcet winner and the sub-optimal arms are larger than the gaps between sub-optimal arms, the bound $K / \min_{i \neq k^*} \Delta_{k^*,i}$ is not improvable (see Section 5).

We introduce and analyze two new procedures. First, we provide in Section 4.1 an algorithm WR-TINF (Weak Regret-Tsallis INF) whose weak regret is bounded by

$$\sqrt{\frac{K}{\min_{i \neq k^*} \Delta_{k^*,i}}} \sqrt{\sum_{i \neq k^*} \frac{1}{\Delta_{k^*,i}}}. \tag{2}$$

This can be upper-bounded by $K / (\min_{i \neq k^*} \Delta_{k^*,i})$. This improves over the state-of-the art [4, 12] (where bounds are respectively of the order of $K^2 / \min_{i \neq j} \Delta_{i,j}^2$ and $K^2 + K / \min_{i \neq k^*} \Delta_{k^*,i}^4$) both in the dependency with respect to $K$ and the gaps. Also, we do not require the strong stochastic transitivity assumption, required in [12][2]. Conversely, the bound $K / \min_{i \neq k^*} \Delta_{k^*,i}$ turns out to be impossible to improve in general –see Section 5.

Second, we introduce in Section 4.2 the algorithm WR-EXP3-IX (Weak Regret EXP3-IX), which, from an heuristic viewpoint aims at eliminating sub-optimal arms by looking at duels between sub-optimal arms. For any $\mathbf{\Delta}$, its weak regret is at most of the order of

$$\sum_{i \neq k^*} \frac{K \log(K / \Delta_*) \Delta_{k^*,i}}{\Delta_{j^*(i),i}^2}, \tag{3}$$

where $j^*(i) \in \arg\max_j \Delta_{j,i}$ and $\Delta_* = \min_{k \neq k^*} \Delta_{k^*,k}$. In the case, where the gaps $\Delta_{j^*(i),i}$ are larger (up to log-terms) than $\Delta_{k^*,i} \sqrt{K}$, the regret guarantee (3) for WR-EXP3-IX becomes smaller than (2) for WR-TINF. Up to our knowledge, WR-EXP3-IX is the first algorithm in weak regret minimization that builds upon the complete structure of the gaps matrix $\mathbf{\Delta}$ to lower the regret.

To further discuss the difference between the performances of both procedures, let us consider a toy model where, for some positive constants, $\Delta_{cw}$ and $\Delta_{sub}$, we have, for any $i \neq k^*$, $\Delta_{k^*,i} = \Delta_{cw}$, that is the gap between the Condorcet winner and the sub-optimal arms. Besides, for any $i \neq k^*$, there exists $j^*(i)$ such that $\Delta_{j^*(i),i} = \Delta_{sub}$. We distinguish two main regimes: (a) If $\Delta_{cw} / \Delta_{sub} \geq 1/\sqrt{K}$, then the weak regret of WR-TINF is the better one and is of the order of $K / \Delta_{cw}$. (b) If $\Delta_{cw} / \Delta_{sub} \leq 1/\sqrt{K}$, then a transition occurs. To show that an arm $k$ is not the Condorcet winner, then it now becomes beneficial to identify arms that provide the most evidence for the suboptimality of $k$. Here, WR-EXP3-IX achieves the better guarantee which is (up to $\log$ terms) of the order of $K^2 (\Delta_{cw} / \Delta_{sub}^2)$.

The presented algorithms use different techniques: we develop WR-TINF using an adaptation of the standard reduction technique (discussed in Section 4.1). We extend the idea of using a best-of-both worlds procedure as a base algorithm to sample each of the two arms $I_t$ and $J_t$. However, since only one of the sampled arms should be optimal, we modify the sampling distribution prescribed by the

---

[2]The strong stochastic transitivity assumption implies that there is a total order between the item, namely if $i$ is preferred to $j$ ($q_{i,j} \geq 1/2$) and $j$ is preferred to $k$ ($q_{j,k} \geq 1/2$), then $q_{i,k} \geq \max(q_{i,j}, q_{j,k})$, see [2].

base algorithm to induce more exploration. The second procedure, WR-EXP3-IX, uses a different approach. Given the value of the left arm $I_t$ (selected in a round-robin manner), we use the EXP3-IX algorithm [11] to select the right arm $J_t$. Then after a fixed number of rounds, the choice of $J_t$ (given the value of $I_t$) concentrates around the arm with highest probability of defeating it. We leverage the fact that when $I_t$ is the Condorcet winner, the gaps are positive, while for sub-optimal arms the minimal gap is negative.

## 4 Upper Bounds

This section presents two algorithms with guarantees on weak regret. Recall that we present two strategies since we identified two regimes as discussed in Section 3.2. Each of the algorithms we present is optimal in one of the regimes and none of them require prior knowledge on the problem parameters.

The first algorithm, WR-TINF, is built upon an adaptation of the reduction technique to a standard multi-armed bandit problem. Its upper bounds depend on the gaps between sub-optimal arms and the Condorcet winner $(\Delta_{k^*,k})_{k\in[K]}$. As demonstrated in the results of Section 5, this algorithm is optimal for some regimes.The second procedure, WR-EXP3-IX, aims for the task of identifying, for each arm, the arm that can eliminate it most rapidly (i.e., the arm with the largest gap). While this strategy results in a quadratic dependence on $K$, we argue that it outperforms WR-TINF for some instances.

### 4.1 Algorithm 1: Weak Regret Tsallis-INF

We adopt a previously explored approach [1, 15, 14], where the dueling bandit problem is converted into two separate multi-armed bandit problems - one for each arm pulled. This reduction was originally applied in the context of strong regret. However, adapting this approach to weak regret requires a more nuanced approach.

The idea of reducing a dueling bandit problem to a standard one was first introduced in [1] where it was termed *Sparring* in the context of minimizing strong regret. The high-level idea of this technique is to view the problem of selecting the the arm pair $(I_t, J_t)$ as two individual multi-armed bandit (MAB) problems. The choice of $I_t$ (resp. $J_t$) can be performed by the first (resp. second) player, following which they incur a loss denoted $\ell_{-1,t}(I_t) := X_t(I_t, J_t)$ (resp. $\ell_{+1,t}(J_t) := 1 - X_t(I_t, J_t)$), where $X_t(I_t, J_t) \sim \text{Ber}(q_{I_t,J_t})$. Here the subscript $-1$ (resp. $+1$) refers to the first (resp. second) player. It is easy to show that the regret of each player $R_{\pm 1,T}$ satisfies the following identity, where $R_T^{(s)}$ is the strong regret of the dueling bandits problem:

$$\mathbb{E}[R_T^{(s)}] = \frac{1}{2}\mathbb{E}[R_{-1,T} + R_{+1,T}]. \tag{4}$$

The last identity reveals that the dueling bandits problem can be addressed using a 'black-box' strategy, where each player is allowed to use a standard Multi-Armed Bandit (MAB) algorithm. In [1], the authors selected the EXP3 algorithm, which provides guarantees suitable for worst-case scenarios. It's important to note that achieving problem-dependent bounds is not possible when the players use stochastic MAB procedures such as Upper Confidence Bounds algorithms, as the losses experienced by the first player, for example, are not stationary. In a later work, [14] implemented a best-of-both-worlds MAB algorithm, specifically the online mirror descent with the Tsallis-INF regularizer [22]. This approach is effective because, from the perspective of the first player, the loss distribution, although variable, is not entirely arbitrary. This is due to the second player's strategy of minimizing their own regret, which involves concentrating on sampling distributions that approximate those associated with the optimal choice, corresponding to the Condorcet winner.

Adopting the reduction above to solve the weak regret dueling bandit problem seems however insufficient due to several reasons. First, Equation (4) shows that minimizing the strong regret, and minimizing the regrets of individual players is equivalent. Second, the weak regret can be significantly smaller than the strong regret. This is because selecting the Condorcet winner just once is sufficient to suffer zero instantaneous weak regret while leaving the second arm free to explore and gain information about the problem. This is not the case in the strong regret minimization where both selected arms have to be Condorcet winners to incur zero instantaneous strong regret. This suggests that the algorithms that are optimal for strong regret cannot be expected to be optimal for weak regret.

For the task of minimizing weak regret, we follow the intuition presented in [14], which involves using a best-of-both-worlds procedure consisting of online mirror descent with the Tsallis-INF regularizer. However, as previously argued, minimizing weak regret necessitates increased exploration, requiring an adjustment to the sampling scheme, denoted as $\boldsymbol{p}_t = (p_{t,i})_{1 \leq i \leq K}$, suggested by these strategies. Our algorithm first performs an internal step where two arms, $I'_t$ and $J'_t$, are sampled independently from the distribution $\boldsymbol{p}_t$ over $[K]$, without playing them. We consider two cases: If $I'_t = J'_t$, we infer that $\boldsymbol{p}_t$ lacks sufficient exploration. In this case, we use the left arm $I_t$ for exploitation by sampling it from $\boldsymbol{p}_t$, and the right arm $J_t$ for exploration. This approach reduces to a decoupled exploration-exploitation problem in standard multi-armed bandits, as studied in [13]. The authors in [13] developed a strategy where exploitation is carried out using $\boldsymbol{p}_t$, while exploration follows a distribution $\boldsymbol{r}_t$ with $r_{t,i} \propto p_{t,i}^{2/3}$. They further demonstrated that this strategy, using Tsallis entropy regularizers with a power of $2/3$, provides instance-specific bounds independent of the time horizon $T$. In the second case, if the internal step results in $I'_t \neq J'_t$, we consider that $\boldsymbol{p}_t$ adequately encourages exploration, and both $I_t$ and $J_t$ are sampled from $\boldsymbol{p}_t$. In summary, given that $I'_t$ and $J'_t$ are sampled from $\boldsymbol{p}_t$, we employ the following strategy:

$$\begin{cases} \text{If } I'_t \neq J'_t : & I_t \sim \boldsymbol{p}_t \text{ and } J_t \sim \boldsymbol{p}_t \\ \text{If } I'_t = J'_t : & I_t \sim \boldsymbol{p}_t \text{ and } J_t \sim \boldsymbol{r}_t, \end{cases} \tag{5}$$

where $\boldsymbol{r}_t = (r_{t,k})_k$ is a distribution over the set $[K]$ defined as follows: for each $k \in [K]$:

$$r_{t,k} := \frac{p_{t,k}^{2/3}}{\sum_{i=1}^{K} p_{t,i}^{2/3}}. \tag{6}$$

Observe that arms with small probability $p_{t,k}$, have a higher chance of being sampled under $\boldsymbol{r}_t$. Hence, the distribution $\boldsymbol{r}_t$ encourages more exploration, which will be beneficial for the weak regret. Finally the losses fed to the online mirror descent procedure are estimated using the importance weight estimators:

$$\hat{\ell}_t(k) := \frac{\mathbb{1}\left(J_t = k\right) X_t(k, I_t)}{q_{t,k}}, \tag{7}$$

where, $q_{t,k} = p_{t,k}$ if $I'_t \neq J'_t$ and $q_{t,k} = r_{t,k}$ if $I'_t = J'_t$. We dedicate Section B in the appendix to develop guarantees on the resulting modified online mirror descent with Tsallis regularizer using the sampling scheme described above.

---

**Algorithm 1** WR-TINF

---

   **Input**: $\alpha$, Learning rates $(\eta_t)$
   **init:** $\hat{L}_0 = 0$.
   **for** $t = 1, \ldots$ **do**
      compute:

$$\boldsymbol{p}_t = \underset{\boldsymbol{p} \in \mathcal{S}^{K-1}}{\arg\min} \left\{ \langle \boldsymbol{p}, \hat{L}_{t-1} \rangle - \frac{1}{\eta_t} \sum_{i=1}^{K} \frac{p_i^{\alpha} - \alpha p_i}{\alpha(1-\alpha)} \right\} \tag{8}$$

      where $\mathcal{S}^{K-1}$ is the $K$-dimensional simplex
      Sample $I'_t$ and $J'_t$ independently following $\boldsymbol{p}_t$
      **if** $I'_t = J'_t$ **then**
         Sample $I_t$ following $\boldsymbol{p}_t$ and $J_t$ following $\boldsymbol{r}_t$ in (6).
      **else**
         Sample $I_t$ and $J_t$ independently following $\boldsymbol{p}_t$
      **end if**
      Play $(I_t, J_t)$, for each $k \in [K]$ compute $\hat{\ell}_t(k)$ using (7) and update: $\hat{L}_t(k) = \hat{L}_{t-1}(k) + \hat{\ell}_t(k)$
   **end for**

---

**Remark 4.1.** *The sampling method used in WR-TINF may occasionally result in selecting the same arm twice ($I_t = J_t$), which is not ideal for weak regret minimization. However, WR-TINF's design ensures that the probability of this event is small enough to maintain the presented guarantees, which are optimal in scenarios that we describe. While we could modify the algorithm to prevent entirely that $I_t = J_t$, such a modification would not enhance our theoretical guarantees significantly.*

**Theorem 4.2.** *Consider Algorithm 1 with $\alpha = 2/3$ and $\eta_t = 2K^{-1/6}/\sqrt{t}$. For any $T \geq 1$, the weak regret satisfies:*

$$\mathbb{E}\left[R_T^{(w)}\right] \leq c\sqrt{\frac{K}{\Delta_*}}\sqrt{\sum_{k \neq k^*} \frac{1}{\Delta_{k^*,k}}},$$

*where $c$ is a numerical constant and $\Delta_* = \min_{k \neq k^*} \Delta_{k^*,k}$.*

We obtain an upper-bound on weak regret of the order of $\tilde{O}(\sqrt{K/\Delta_*}\sqrt{\sum_{k \neq k^*} 1/\Delta_{k^*,k}})$. In the setting where all the gaps for the Condorcet winner are constant, the upper bound above translates to $\mathcal{O}(K/\Delta_{\mathrm{cw}})$. The last optimality result is shown in Theorem 5.1.

### 4.2 Algorithm 2: Weak Regret-EXP3-IX

This algorithm uses the Implicit Exploration strategy (EXP3-IX) [11], which is adapted specifically for the dueling bandits problem. At its core, the algorithm selects one arm (left arm $I_t$) for exploitation and another (right arm $J_t$) for exploration. Given $I_t$, $J_t$ is selected following the EXP3-IX procedure. We chose the EXP3-IX algorithm (restated in Section D.2 of the Appendix), particularly the version without a fixed horizon for technical reasons, namely its bounds on cumulative loss that hold with high probability. To clarify the notation used: in each round, we observe the result of the duel between $I_t$ and $J_t$, denoted by $X_t(I_t, J_t)$. The variable $X_t(i, j)$ represents the duel outcome between arm $i$ and arm $j$ in round $t$.

The algorithm operates across multiple stages, where each stage $n \geq 1$ is defined by a threshold value $B = 2^{n-1}$, updated through a doubling technique. At every stage, given $B$, we consistently select the left arm as $I_t = i$, and consider a standard Multi-Armed Bandit problem where the choices are the duels between arm $i$ and the other arms in $[K] \setminus i$. Specifically, these choices relate to the variables $X_t(i, j) - \frac{1}{2} : j \in [K] \setminus i$. Recall that $\mathbb{E}\left[X_t(i, j) - \frac{1}{2}\right] = \Delta_{i,j}$, and the optimal arm, which minimizes cumulative loss, is $j^*(i) = \arg\min_{j \in [K] \setminus i} \Delta_{i,j}$. The cumulative loss after executing EXP3-IX for this specified problem over $u$ rounds is denoted by $S(i, n, u)$.

$$S(i, n, u) := \sum_{s=\tau+1}^{u+\tau} \left(X_s(i, J_s) - \frac{1}{2}\right),$$

where $\tau$ represents the round at which the procedure starts. We continue the procedure until the value of $S(i, n, u)$ reaches the threshold $-B\sqrt{u}$. When this threshold is met, we transition to the next arm, $i + 1$, and address the duels involving this new arm. A stage is completed once all arms have met this stopping criterion, allowing the algorithm to advance to the next stage, $n + 1$.

The underlying rationale of the algorithm is as follows: consider stage $n$, by design of the algorithm, if $i$ is a sub-optimal arm, then after a constant number of rounds, the process $S(i, n, u)$ mimics a random walk characterized by a negative drift of $\Delta_{i,j^*(i)} < 0$. We demonstrate that $S(i, n, u)$ typically reaches the threshold $-B\sqrt{u}$ when $u$ is approximately of the order $\max\{K, B\}/\Delta_{j^*(i),i}^2$. In contrast, the process $S(k^*, n, u)$, which is linked to the Condorcet winner, has a positive drift. We show that the probability of the last process never meeting the threshold $-B\sqrt{u}$ for some $u \geq 1$ is less than $\min\{1, \exp(-B^2)\log(1/\Delta_*)\}$, where $\Delta_* = \min_{k \neq k^*} \Delta_{k^*,k}$. Consequently, there is a high probability that, at some stage, the algorithm will be trapped in a loop where the left arm is the Condorcet winner leading to zero regret when considering weak regret.

**Theorem 4.3.** *Under the assumption of the existence of a Condorcet winner, the weak regret of Algorithm 2 satisfies:*

$$\mathbb{E}\left[R_T^{(w)}\right] \leq c\log(K/\Delta_*)\sum_{k \neq k^*} \frac{K\Delta_{k^*,k}}{\Delta_{j^*(k),k}^2},$$

*where for each $k \neq k^*$: $j^*(k) \in \arg\max_j \Delta_{j,k}$, $\Delta_* = \min_{k \neq k^*} \Delta_{k^*,k}$ and $c = c'\max\{1, \log\log\log(K \vee 16)\}$ with $c'$ being an absolute constant.*

The complete proof is presented in Section D of the supplementary material.

*Comparison with WR-TINF:* Intuitively, WR-EXP3-IX is designed to outperform WR-TINF when the gaps between sub-optimal arms are more important than the gaps with the Condorcet winner. In

---
**Algorithm 2** WR-EXP3-IX
---
**Initialization:** $B \leftarrow 1$ and for all $i$: $n_i \leftarrow 0$, $S(i) \leftarrow 0$.
**for** $t = 1, \dots$ **do**
    **for** $i = 1, \dots, K$ **do**
        **while** $S(i) > -B\sqrt{n_i}$ **do**
            Run EXP3-IX algorithm on the problem with variables $\{X(i,k) - \frac{1}{2} : k \in [K] \setminus \{i\}\}$,
            after each round, update the cumulative loss $S(i)$ and the number of rounds $n_i$.
        **end while**
        Re-initialize: $S(i) \leftarrow 0$, $n_i \leftarrow 0$.
    **end for**
    $B \leftarrow 2B$.
**end for**
---

this case, as argued in Section 3.2, the algorithm should explore the $K^2$ duels to detect sub-optimal arms. More rigorously, consider an example where the gaps satisfy: For all $i \neq k^*$: $\Delta_{k^*,i} = \Delta_{\text{cw}}$ and $\max_{j \neq k^*} \Delta_{j,i} = \Delta_{\text{sub}}$, where $\Delta_{\text{cw}}$ and $\Delta_{\text{sub}}$ are positive constants. Then the upper bound in Theorem 4.3 is of order $K^2 \Delta_{\text{cw}} / \Delta_{\text{sub}}^2$. The last bound is sharper than the bound for WR-TINF (which is of order $K / \Delta_{\text{cw}}$), when we have $\Delta_{\text{sub}} / \Delta_{\text{cw}} > 1/\sqrt{K}$.

## 5 Lower Bound

In this section, we provide a lower bound on the largest weak regret of any algorithm, when confronted with a given set of dueling bandit problems, which we will discuss below.

Let $\Delta_{\text{cw}} \in (0, 1/4)$ denote a positive number. For a dueling bandits problem, define the class of problems $\mathbb{D}(\Delta_{\text{cw}})$ by the set of matrices $M$ representing the gaps $(\Delta_{i,j})_{ij}$ such that $M$ is skew-symmetric and there exists some $k^* \in [K]$ (representing the Condorcet winner) such that:

$$\forall i \neq k^* : M_{k^*,i} = \Delta_{\text{cw}} \text{ and } \forall i, j \neq k^* : |M_{i,j}| \leq \Delta_{\text{cw}}.$$

The introduced class of matrices $\mathbb{D}(\Delta_{\text{cw}})$ includes many natural instances, such as when the gaps satisfy the general identifiability assumption. This assumption states that for each sub-optimal arm $j$, the arm with the highest probability to beat $j$ is the Condorcet winner $k^*$: i.e., $k^* \in \arg\min_{i \in [K]} \Delta_{j,i}$. It has been considered in prior works such as [21] and more specifically it is implied by strong stochastic transitivity assumption (Section 3.1 of [2]).

**Theorem 5.1.** *Fix $K \geq 6$, $\Delta_{cw} \in (0, 1/4)$. The weak regret of an algorithm $\mathcal{A}$ satisfies:*

$$\max_{M \in \mathbb{D}(\Delta_{cw})} \mathbb{E}_{M,\mathcal{A}}[R_T] \geq c \frac{K}{\Delta_{cw}},$$

*when $T \geq c' K / \Delta_{cw}^2$. Here $c$ and $c'$ are numerical constants.*

The result in Theorem 5.1 proves that Algorithm 1 is optimal for the considered instance, particularly highlighting that linear scaling with $K$ is optimal in this case. In the lower bound, we assumed uniform gaps between the Condorcet winner and the sub-optimal arms (equal to $\Delta_{\text{cw}}$). A potential improvement would be to develop a lower bound that depends on all the gaps with the Condorcet winner $(\Delta_{k^*,i})_{i \in [K]}$. Additionally, a more general lower bound should discard the general identifiability assumption. As previously argued, if the gaps between sub-optimal arms are large compared to the gaps with the Condorcet winner, it becomes easier to explore the $K^2$ duels to detect the sub-optimality of the arms and focus decision-making on the Condorcet winner.

## 6 Experiments

In this section, we perform a numerical evaluation of WR-EXP3-IX and WR-TINF algorithms in three different scenarios that favor different algorithms according to the prior theoretical results. As a benchmark for our experiments, we utilize the state-of-the-art algorithm for weak regret, WS-W [4]. Additionally, we include one of the best-performing algorithms for strong regret, Versatile-DB

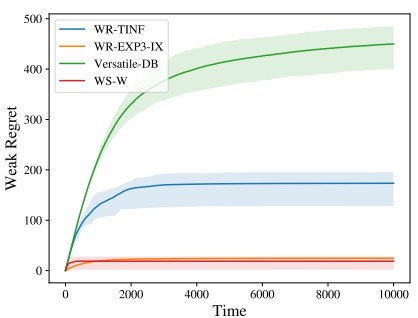

(a) Weak regret for small problem (Scenario 1)

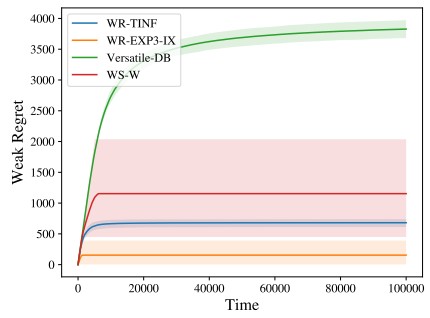

(b) Weak regret for moderate problem (Scenario 2)

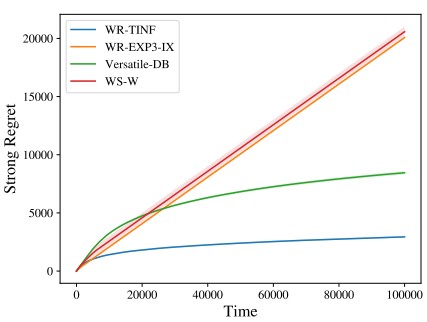

(c) Strong regret for moderate problem (Scenario 2)

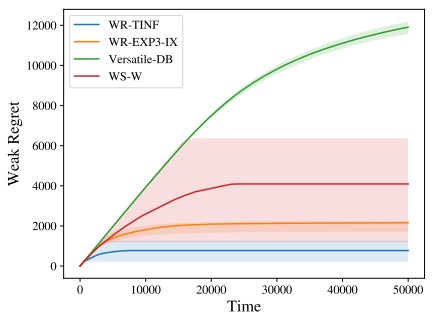

(d) Weak regret for large problem (Scenario 3)

Figure 1: Performance of algorithms in different scenarios

[14], to demonstrate that optimizing for strong regret does not necessarily translate into optimal weak regret performance. For each of the experiments, we plot the mean regret over 20 iterations together with $0.2$ and $0.8$ quantiles. All the experiments in this section use theoretical values of parameters for the algorithms. The runtime of each algorithm and iteration is in terms of minutes on a personal computer.

**Scenario 1: weak regret under SST (Figure 1a).** We consider here: $K = 30$, $T = 10000$, $q_{i,j} = 1 - q_{j,k^*} = 0.8$ for $i, j \in [K]$ such that $i < j$. In this scenario, we have a small number of arms and the SST holds - the arm with the lower index always wins with probability $0.8$. This favors WS-W and WR-EXP3-IX algorithms. On the other hand, WR-TINF is a explores less, this results in larger regret for small $K$ while the algorithm shines as $K$ grows.

**Scenario 2: Strong and weak regret comparison without SST (Figures 1b and 1c).** We consider here: $K = 150$, $T = 100000$, $q_{k^*,i} = 0.9$ for every $i \in [K] \setminus \{k^*\}$, $q_{i,j} = 0.9$ (resp. $q_{i,j} = 0.1$) for $i, j \in [K] \setminus \{k^*\}$ such that $i < j$ and $(i + j) \equiv 0 \mod 2$ (resp. $(i + j) \equiv 1 \mod 2$). In this scenario, we have a moderately large number of arms and SST does not hold - each arm, except for the Condorcet winner, wins against approximately $K/2$ (every other index) other arms with probability $0.9$ and loses to the other arms with probability $0.1$. This should still favor WR-EXP3-IX algorithm but lack of ordering makes WS-W algorithm perform slightly worse. Algorithm WR-TINF slightly closes the gap in weak regret thanks to the increased number of arms. This can be seen in Figure 1b. For completeness of comparison, we also plot strong regret of the algorithms, see Figure 1c. Naturally, algorithms WS-W and WR-EXP3-IX suffer linear strong regret since they never play the same arm twice. However, WR-TINF performs well even with extra exploration, needed for weak regret, compared to Versatile-DB.

**Scenario 3: large number of arms, no SST (Figure 1d).** We consider: $K = 400$, $T = 50000$, $q_{k^*,i} = 0.9$ for every $i \in [K] \setminus \{k^*\}$, $q_{i,j} = 0.9$ (resp. $q_{i,j} = 0.1$) for $i, j \in [K] \setminus \{k^*\}$ such that $i < j$ and $(i + j) \equiv 0 \mod 2$ (resp. $(i + j) \equiv 1 \mod 2$). The same setup without SST as in Scenario 2 but with a larger $K$. Better scaling with $K$ gives algorithm WR-TINF an edge over algorithms WS-W and WR-EXP3-IX while WS-W still suffers from the lack of SST.

**Remark 6.1.** *On the variance of the WS-W algorithm: WS-W is a round-based procedure where the selected arms, "winner and challenger," duel in batches of iterations. The length of each batch increases with the number of duels won by the selected arms so far. When an arm loses, it is replaced by a contender chosen from the remaining arms. Once the set of candidate arms is exhausted, the process is repeated. In numerical experiments, particularly with a large number of arms (Scenario 3 in the simulations section), we observe that in some unfortunate cases, especially in the early stages, the CW may lose its duels. This results in a large number of iterations before it is picked again as a contender, leading to very high weak regret for the procedure. Although such outcomes are infrequent, they significantly impact the empirical variance of the weak regret of WS-W.*

## 7 Conclusion and limitations

In this work, we addressed the problem of weak regret analysis under the assumption of a Condorcet winner. We showed that, it is impossible in general to achieve a weak regret smaller than $K/(\min_{i \neq k^*} \Delta_{k^* i})$ and we introduced the procedure WR-TINF which achieves this bound. The second algorithm, WR-EXP3-IX, employs a more aggressive exploration strategy by querying the $K^2$ duels. We show that in some cases, this approach, despite inducing a quadratic dependence on $K$ can outperform WR-TINF, because it better adapts to the gaps between suboptimal arms. This is the first work in duelling bandit with weak regret that establishes how that the full matrix $\mathbf{\Delta}$ can be leveraged in the regret.

This work gives rise to several open questions. While WR-TINF is optimal in certain instances, developing algorithms that fully adapt to the underlying problem parameters remains a significant challenge.

## Acknowledgments and Disclosure of Funding

The work of E.M. Saad and N. Verzelen has been partially supported by grant ANR-21-CE23-0035 (ASCAI,ANR). The work of A. Carpentier is partially supported by the Deutsche Forschungsgemeinschaft (DFG)- Project-ID 318763901 - SFB1294 "Data Assimilation", Project A03, by the DFG on the Forschungsgruppe FOR5381 "Mathematical Statistics in the Information Age - Statistical Efficiency and Computational Tractability", Project TP 02 (Project-ID 460867398), by the Agence Nationale de la Recherche (ANR) and the DFG on the French-German PRCI ANR-DFG ASCAI CA1488/4-1 "Aktive und Batch-Segmentierung, Clustering und Seriation: Grundlagen der KI" (Project-ID 490860858).

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

# 8 Appendix / supplemental material

## A  Preliminary results for Algorithm 1

### A.1  Preliminary Lemma:

Lemma below upper-bounds the weak regret of Algorithm 2.

**Lemma A.1.** *Let an algorithm $\mathcal{A}'$ playing in each round the pair of arms $(I'_t, J'_t)$ observing only the feedback of the duel between arms $(I_t, J_t)$ sampled following the scheme in (5). Let $\mathcal{A}$ be an algorithm playing in each round the pair $(I_t, J_t)$ and observing the feedback $X_t(I_t, J_t)$. let $R^{(s)}_{T,\mathcal{A}'}$ denote the strong regret of $\mathcal{A}'$ and $R^{(w)}_{T,\mathcal{A}}$ the weak regret of $\mathcal{A}$. We have:*

$$\mathbb{E}\left[R^{(w)}_{T,\mathcal{A}}\right] \leq \mathbb{E}\left[R^{(s)}_{T,\mathcal{A}'}\right].$$

*Proof.* For $t \in [T]$ observe that following Algorithm 1, we have:

$$
\begin{aligned}
\mathbb{E}\left[\min\{\Delta_{k^*,I_t}; \Delta_{k^*,J_t}\}\right] &= \mathbb{P}\left(I'_t \neq J'_t\right) \mathbb{E}\left[\min\{\Delta_{k^*,I_t}; \Delta_{k^*,J_t}\}\mid I'_t \neq J'_t\right] \\
&\quad + \mathbb{P}\left(I'_t = J'_t\right) \mathbb{E}\left[\min\{\Delta_{k^*,I_t}; \Delta_{k^*,J_t}\}\mid I'_t = J'_t\right] \\
&\leq \mathbb{P}\left(I'_t \neq J'_t\right) \mathbb{E}\left[\min\{\Delta_{k^*,I'_t}; \Delta_{k^*,J'_t}\}\right] + \mathbb{P}\left(I'_t = J'_t\right) \mathbb{E}\left[\Delta_{k^*,I'_t}\right] \\
&\leq \mathbb{P}\left(I'_t \neq J'_t\right) \mathbb{E}\left[\frac{\Delta_{k^*,I'_t} + \Delta_{k^*,J'_t}}{2}\right] + \mathbb{P}\left(I'_t = J'_t\right) \mathbb{E}\left[\Delta_{k^*,I'_t}\right] \\
&= \mathbb{E}\left[\frac{\Delta_{k^*,I'_t} + \Delta_{k^*,J'_t}}{2}\right].
\end{aligned}
$$

We used in the second line the fact that conditionally to the event $\{I'_t \neq J'_t\}$, $(I_t, J_t)$ are sampled independently from $\boldsymbol{p}_t$ (same distribution as $(I'_t, J'_t)$), and conditionally to the event $\{I'_t = J'_t\}$, $I_t$ is sampled from $\boldsymbol{p}_t$. The result follows by summing over $t \in [T]$. $\qquad\square$

### A.2  Additional Lemmas:

Lemma below states that the reduction proved in Theorem 2 of [14] is still valid in our setting. Recall the notation:

$$R'_{-1,T} = \sum_{t=1}^{T} \ell'_{-1,t}(I'_t) - \ell'_{-1,t}(k^*)$$

$$R'_{+1,T} = \sum_{t=1}^{T} \ell'_{+1,t}(J'_t) - \ell'_{+1,t}(k^*),$$

where $\ell'_{-1,t}(k) = X_t(J'_t, k)$ and $\ell'_{+1,t}(k) = X_t(I'_t, k)$.

**Lemma A.2.** *Theorem 2 of [14] The expected strong regret of algorithm $\mathcal{A}'$ satisfies:*

$$
\begin{aligned}
\mathbb{E}[R^{(s)}_{T,\mathcal{A}'}] &= \frac{1}{2}\mathbb{E}\left[R'_{-1,T} + R'_{+1,T}\right] \\
&= \mathbb{E}\left[R'_{-1,T}\right].
\end{aligned}
$$

*Moreover, we have:*

$$\mathbb{E}\left[R'_{-1,T}\right] = \sum_{t=1}^{T} \mathbb{E}\left[\sum_{k \neq k^*} p_{t,k} \Delta_{k^*,k}\right].$$

*Proof.* We have

$$\mathbb{E}\left[R'_{-1,T} + R'_{+1,T}\right] = \sum_{t=1}^{T} \mathbb{E}\left[\ell'_{-1,t}(I'_t) - \ell'_{-1,t}(k^*) + \ell'_{+1,t}(J'_t) - \ell'_{+1,t}(k^*)\right]$$

$$= \sum_{t=1}^{T} \mathbb{E}\left[1 - \left(\ell'_{-1,t}(k^*) + \ell'_{+1,t}(k^*)\right)\right]$$

$$= \sum_{t=1}^{T} \mathbb{E}\left[\Delta_{k^*,I'_t} + \Delta_{k^*,J'_t}\right] \qquad (9)$$

$$= 2\mathbb{E}\left[R^{(s)}_{T,\mathcal{A}'}\right].$$

We used in the second line the fact that: $\ell'_{-1,t} + \ell'_{+1,t} = X_t(I'_t, J'_t) + X_t(J'_t, I'_t) = 1$.

Furthermore, observe that for any $t \in [T]$:

$$\mathbb{E}\left[\ell'_{-1,t}(I'_t) - \ell'_{-1,t}(k^*)\right] = \mathbb{E}\left[X_t(I'_t, J'_t) - X_t(k^*, J'_t)\right]$$

$$= \mathbb{E}\left[X_t(J'_t, I'_t) - X_t(k^*, I'_t)\right]$$

$$= \mathbb{E}\left[\ell'_{+1,t}(J'_t) - \ell'_{+1,t}(k^*)\right],$$

where we used the fact that $I'_t$ and $J'_t$ are sampled independently from the same distribution. Therefore we have summing over $t$:

$$\mathbb{E}\left[R'_{-1,T}\right] = \mathbb{E}\left[R'_{+1,T}\right]$$

Now let us prove the last identity of the lemma. We have

$$\mathbb{E}\left[R'_{-1,T}\right] = \frac{1}{2}\mathbb{E}\left[R'_{-1,T} + R'_{+1,T}\right]$$

$$= \frac{1}{2}\sum_{t=1}^{T} \mathbb{E}\left[\Delta_{k^*,I'_t} + \Delta_{k^*,J'_t}\right]$$

$$= \sum_{t=1}^{T} \mathbb{E}\left[\Delta_{k^*,I'_t}\right]$$

$$= \sum_{t=1}^{T} \sum_{k \neq k^*} \mathbb{E}\left[p_{k,t}\Delta_{k^*,k}\right].$$

We used in the last equality (9), and the fact that $I'_t \sim J'_t$ in the fourth. $\qquad \square$

Recall the notation: $\ell'_t(k) := X_t(k, J'_t)$, which corresponds to the loss of the learner playing $k$ when the environment chooses $J'_t$. Recall the expression of the importance weights estimator introduced in (7) in the main body:

$$\hat{\ell}_t(k) := \frac{\mathbb{1}\left(J_t = k\right) X_t(k, I_t)}{q_{t,k}},$$

where,

$$q_{t,k} = \left(\frac{\mathbb{1}\left(I'_t \neq J'_t\right)}{\mathbb{P}\left(J_t = k \mid I'_t \neq J'_t\right)} + \frac{\mathbb{1}\left(I'_t = J'_t\right)}{\mathbb{P}\left(J_t = k \mid I'_t = J'_t\right)}\right)^{-1}$$

We provide the following expressions of the probabilities used in the definition above. we have that $I'_t$ and $J'_t$ are sampled independently following $\boldsymbol{p}_t = (p_{t,k})_{k \in [K]}$:

$$\mathbb{P}\left(J_t = k \mid I'_t \neq J'_t\right) = p_{t,k}$$

$$\mathbb{P}\left(J_t = k \mid I'_t = J'_t\right) = \frac{p_{t,k}^{2/3}}{\sum_{i=1}^{K} p_{t,i}^{2/3}}.$$

**Lemma A.3.** *We have for any $k \in [K]$:*

$$\mathbb{E}_{t-1}[\hat{\ell}_t(k)] = \mathbb{E}_{t-1}\left[\ell'_t(k)\right],$$

*where $\mathbb{E}_{t-1}[.]$ corresponds to the expectation at round $t$ given the past information.*

*Proof.* We have

$$\mathbb{E}_{t-1}[\hat{\ell}_t(k)] = \underbrace{\mathbb{E}_{t-1}\left[\frac{\mathbb{1}\left(J_t = k\right)\mathbb{1}\left(I'_t \neq J'_t\right)X_t(k, I_t)}{\mathbb{P}\left(J_t = k \mid I'_t \neq J'_t\right)}\right]}_{\text{Term 1}} + \underbrace{\mathbb{E}_{t-1}\left[\frac{\mathbb{1}\left(J_t = k\right)\mathbb{1}\left(I'_t = J'_t\right)X_t(k, I_t)}{\mathbb{P}\left(J_t = k \mid I'_t = J'_t\right)}\right]}_{\text{Term 2}}$$

**Calculating Term 1:** We have

$$\text{Term 1} = \mathbb{E}_{t-1}\left[\frac{\mathbb{1}\left(J_t = k\right)\mathbb{1}\left(I'_t \neq J'_t\right)X_t(k, I_t)}{\mathbb{P}\left(J_t = k \mid I'_t \neq J'_t\right)}\right]$$

$$= \mathbb{P}\left(I'_t \neq J'_t\right)\mathbb{E}_{t-1}\left[\frac{\mathbb{1}\left(J_t = k\right)X_t(k, I_t)}{\mathbb{P}\left(J_t = k \mid I'_t \neq J'_t\right)} \mid I'_t \neq J'_t\right]$$

$$= \mathbb{P}\left(I'_t \neq J'_t\right)\mathbb{E}_{t-1}\left[\frac{\mathbb{1}\left(J_t = k\right)X_t(k, J'_t)}{\mathbb{P}\left(J_t = k \mid I'_t \neq J'_t\right)} \mid I'_t \neq J'_t\right]$$

$$= \mathbb{P}\left(I'_t \neq J'_t\right)\mathbb{E}_{t-1}[\hat{\ell}_t(k) \mid I'_t \neq J'_t],$$

where we used in the third line the fact that conditionally to $I'_t \neq J'_t$, we have $I_t \sim J'_t$.

**Calculating Term 2:** We have

$$\text{Term 2} = \mathbb{E}_{t-1}\left[\frac{\mathbb{1}\left(J_t = k\right)\mathbb{1}\left(I'_t = J'_t\right)X_t(k, I_t)}{\mathbb{P}\left(J_t = k \mid I'_t = J'_t\right)}\right]$$

$$= \mathbb{P}\left(I'_t = J'_t\right)\mathbb{E}_{t-1}\left[\frac{\mathbb{1}\left(J_t = k\right)X_t(k, I_t)}{\mathbb{P}\left(J_t = k \mid I'_t = J'_t\right)} \mid I'_t = J'_t\right]$$

$$= \mathbb{P}\left(I'_t = J'_t\right)\mathbb{E}_{t-1}\left[\frac{\mathbb{1}\left(J_t = k\right)X_t(k, J'_t)}{\mathbb{P}\left(J_t = k \mid I'_t = J'_t\right)} \mid I'_t = J'_t\right]$$

$$= \mathbb{P}\left(I'_t = J'_t\right)\mathbb{E}_{t-1}[\hat{\ell}_t(k) \mid I'_t = J'_t].$$

The conclusion follows by summing the obtained expressions. $\qquad\square$

# B  Analysis for the modified OMD with Tsallis regularizer

## B.1  The Setting:

The online mirror descent with Tsallis regularizer in the standard coupled exploration and exploitation case was analyzed in [22] and in the decoupled exploration and exploitation setting in [13]. In this section, we develop guarantees in the case where exploration and exploitation are partially coupled via the sampling scheme that we employ, which is restated in Algorithm 4. Note that to be compatible with our setting for dueling bandits, we need to make some modifications to the game protocol for the problem of regret minimization. More precisely, we assume that in each round $t$ instead of choosing a sequence of numbers, the environment chooses a sequence of distributions for losses. The incurred and observed losses are sampled independently from the sequence chosen by the environment. Note that this change doesn't affect the definition of the pseudo-regret, which is the quantity of interest here, since the definitions involve expectations. We present in Algorithm 3 the game protocol of this game.

---

**Algorithm 3** Game Protocol

---

**for** $t = 1, \ldots$ **do**

    The player chooses two distributions $\boldsymbol{p}_t = (p_{t,k})_{k \in [K]}$ and $\boldsymbol{q}_t = (q_{t,k})_{k \in [K]}$ over $[K]$.

    Concurrently, the environment chooses $K$ random distributions with support in $[0,1]$ denoted $(\mathcal{L}_{t,k})_{k \in [K]}$.

    The player plays an arm $A_t$ sampled following $\boldsymbol{p}_t$ and incurs an unseen loss $\ell_{t,A_t}$ sampled from $\mathcal{L}_{t,A_t}$.

    The player samples an arm $B_t$ following $\boldsymbol{q}_t$ and observes an independent fresh sample $\bar{\ell}_{t,B_t}$ from the distribution $\mathcal{L}_{t,B_t}$.

**end for**

---

**Remark B.1.** *The only difference between the game protocol above and the one used in [13] is that in our setting we assume that the observed losses and the incurred losses are independently sampled from some distribution, while in [13] the losses that are observed and incurred are the same.*

We define the pseudo-regret with respect to $k$ as follows:

$$\mathcal{R}_T := \sum_{t=1}^{T} \mathbb{E}\left[\ell_{t,A_t}\right] - \min_k \mathbb{E}\left[\sum_{t=1}^{T} \ell_{t,k}\right].$$

We denote by $k^* := \arg\min_k \mathbb{E}\left[\sum_{t=1}^{T} \ell_{t,k}\right]$.

We consider the stochastically constrained adversarial setting where we have for some positive numbers $(\Delta_k)$:

$$\mathcal{R}_T = \sum_{t=1}^{T} \sum_{k \neq k^*} \mathbb{E}\left[p_{t,k}\right] \Delta_k. \tag{10}$$

## B.2 The Algorithm:

Following [22], we consider a follow-the-regularized leader (FTRL) approach using

$$\Psi_t(w) = -\frac{1}{\eta_t} \sum_{k=1}^{K} \frac{w_k^\alpha - \alpha w_k}{\alpha(1-\alpha)},$$

as regularizers where $(\eta_t)$ is a sequence of positive numbers and $\alpha \in (0,1)$.

More specifically, following the analysis of decoupled exploration and exploitation in [13], we focus on the case where $\alpha = 2/3$. We introduce the following loss estimators:

$$\forall t \in [T], k \in [K], \; \hat{\ell}_{t,k} = \frac{\mathbb{1}\left(B_t = k\right)}{q_{t,k}} \, \bar{\ell}_{t,k}, \tag{11}$$

Recall that following Game Protocol 3: $\bar{\ell}_{t,k}$ and $\ell_{t,k}$ are independent and follow the same distribution $\mathcal{L}_{t,k}$.

We define our exploration distribution $\boldsymbol{q}_t$ by:

$$\frac{\mathbb{1}\left(B_t = k\right)}{q_{t,k}} := \frac{\mathbb{1}\left(\{B_t = k\} \text{ and } E_t\right)}{q_{t,k}^{(1)}} + \frac{\mathbb{1}\left(\{B_t = k\} \text{ and } E_t^c\right)}{q_{t,k}^{(2)}}, \tag{12}$$

where $E_t$ is some internal event specified by the learner based on past information and some internal randomization. In Algorithm 4 below, $E_t$ corresponds to the event $\{A_t = B_t'\}$ where $A_t$ and $B_t'$ are independent and sampled from $[K]$ following $\boldsymbol{p}_t$. $E_t^c$ denotes the complementary to $E_t$. $(q_{t,k}^{(1)})_{k \in [K]}$ and $(q_{t,k}^{(2)})_{k \in [K]}$ are probability distributions defined by:

$$q_{t,k}^{(1)} = \mathbb{P}\left(B_t = k \mid E_t\right)$$

$$q_{t,k}^{(2)} = \mathbb{P}\left(B_t = k \mid E_t^c\right).$$

We define the probability $(q_{t,k}^{(1)})_{k \in [K]}$ by: $q_{t,k}^{(1)} := p_{t,k}$. Define the probability $q_{t,k}^{(2)}$ as follows:

$$q_{t,k}^{(2)} := \frac{p_{t,k}^{2/3}}{\sum_{i=1}^{K} p_{t,i}^{2/3}}. \tag{13}$$

---

**Algorithm 4** Partially coupled Tsallis-INF

---

**Input**: $(\Psi_t)_{t=1,2,\dots}$
**init:** $\hat{L}_0 = 0$.
**for** $t = 1, \dots$ **do**
    choose $\boldsymbol{p}_t = \arg\max_p \left\{ \langle p, -\hat{L}_{t-1} \rangle - \Psi_t(p) \right\}$
    Sample $A_t$ from $[K]$ using $\boldsymbol{p}_t$.
    Play $A_t$ and suffer $\ell_{t,A_t}$.
    Sample $B_t'$ independently from $[K]$ using $\boldsymbol{p}_t$.
    **if** $B_t' \neq A_t$ **then**
        Sample $B_t$ according to $\boldsymbol{p}_t$.
    **else**
        Sample $B_t$ according to $(q_{t,k}^{(2)})$ defined in (13).
    **end if**
    Observe $\bar{\ell}_{t,B_t}$ (loss having the same distribution as $\ell_{t,B_t}$)
    Compute $\hat{\ell}_{t,k}$ using (11) and update $\hat{L}_t$.
**end for**

---

**Theorem B.2.** *Suppose the regret satisfies the self-constraining condition* (10). *The pseudo-regret of Algorithm 4 with* $\alpha = 2/3$, $\eta_t = \frac{2K^{-1/6}}{\sqrt{t}}$, $\Psi_t(w) = -\frac{1}{\eta_t} \sum_i \frac{w_i^\alpha - \alpha w_i}{\alpha(1-\alpha)}$, *satisfies:*

$$\mathcal{R}_T \leq c \sqrt{\frac{K}{\Delta_*}} \sqrt{\sum_{k \neq k^*} \frac{1}{\Delta_k}}$$

*where* $\Delta_* = \min_{k \neq k^*} \Delta_k$ *and $c$ is a numerical constant.*

*Proof.* Following previous works, we decompose the expected regret into the stability and penalty terms using the potential $\Phi_t$ defined by:

$$\Phi_t(-L) = \max_{w \in \mathcal{S}^{K-1}} \left\{ \langle w, -L \rangle + \frac{1}{\eta_t} \sum_{k=1}^{K} \frac{w_k^\alpha - \alpha w_k}{\alpha(1-\alpha)} \right\},$$

where $\mathcal{S}^{K-1}$ is the set of probability weights on $[K]$. Let $\hat{L}_t = \sum_{t=1}^{T} \hat{\ell}_t$, where $\hat{\ell}_t$ is defined by (11). We have

$$\mathcal{R}_T = \mathbb{E}\underbrace{\left[ \sum_{t=1}^{T} \ell_{t,A_t} + \Phi_t(-\hat{L}_t) - \Phi_t(-\hat{L}_{t-1}) \right]}_{\text{stability}} + \mathbb{E}\underbrace{\left[ \sum_{t=1}^{T} \Phi_t(-\hat{L}_{t-1}) - \Phi_t(-\hat{L}_t) - \ell_{k^*,t} \right]}_{\text{penalty}}.$$

Recall that we have:

$$\mathcal{R}_T = \sum_{t=1}^{T} \sum_{k \neq k^*} \mathbb{E}[p_{t,k}] \Delta_k.$$

Therefore:

$$\mathcal{R}_T = 2\mathcal{R}_T - \sum_{t=1}^{T}\sum_{k\neq k^*}\mathbb{E}[p_{t,k}]\Delta_k$$

$$= \underbrace{2\mathbb{E}\left[\sum_{t=1}^{T}\ell_{t,A_t} + \Phi_t(-\hat{L}_t) - \Phi_t(-\hat{L}_{t-1})\right] - \frac{1}{2}\sum_{t=1}^{T}\sum_{k\neq k^*}\mathbb{E}[p_{t,k}]\Delta_k}_{\text{Term 1}}$$

$$+ \underbrace{2\mathbb{E}\left[\sum_{t=1}^{T}\Phi_t(-\hat{L}_{t-1}) - \Phi_t(-\hat{L}_t) - \ell_{t,k^*}\right] - \frac{1}{2}\sum_{t=1}^{T}\sum_{k\neq k^*}\mathbb{E}[p_{t,k}]\Delta_k}_{\text{Term 2}}.$$

Let us bound each term separately.

**Bounding the term corresponding to the penalty: Term 2**    Let $T_0 \geq 1$. We have using Lemma B.4

$$\text{Term 2} \leq \frac{9\sqrt{K}}{2} + \sum_{t=1}^{T}\sum_{i\neq k^*}\left(\frac{9}{4}K^{1/6}\frac{\mathbb{E}\left[p_{t,i}\right]^{2/3}}{\sqrt{t}} - \frac{1}{2}\mathbb{E}\left[p_{t,i}\right]\Delta_i\right) \tag{14}$$

$$\leq \frac{9\sqrt{K}}{2} + \sum_{t=1}^{T_0}\sum_{i\neq k^*}\left(\frac{9}{4}K^{1/6}\frac{\mathbb{E}\left[p_{t,i}\right]^{2/3}}{\sqrt{t}}\right) + \sum_{t=T_0+1}^{T}\sum_{i\neq k^*}\left(\frac{9}{4}K^{1/6}\frac{\mathbb{E}\left[p_{t,i}\right]^{2/3}}{\sqrt{t}} - \frac{1}{2}\mathbb{E}\left[p_{t,i}\right]\Delta_i\right)$$

$$\leq \frac{9\sqrt{K}}{2} + \frac{9}{2}\sqrt{KT_0} + \sum_{t=T_0+1}^{T}\sum_{i\neq k^*}\left(\frac{9}{4}K^{1/6}\frac{\mathbb{E}\left[p_{t,i}\right]^{2/3}}{\sqrt{t}} - \frac{1}{2}\mathbb{E}\left[p_{t,i}\right]\Delta_i\right)$$

$$\leq 9\sqrt{KT_0} + \sum_{t=T_0+1}^{T}\sum_{i\neq k^*}\max_{z\geq 0}\left\{\frac{9K^{1/6}z^{2/3}}{4\sqrt{t}} - \frac{\Delta_i}{2}z\right\}, \tag{15}$$

where we used in the third line Jensen's inequality on the concave function $x \to x^{2/3}$, giving: $\sum_{i\neq k^*}\mathbb{E}[p_{t,i}^{2/3}] \leq K^{1/3}$ and the fact that $\sum_{t=1}^{T_0}\frac{1}{\sqrt{t}} \leq 2\sqrt{T_0}$.

**Bounding the term corresponding to the stability: Term 1**    We have using Lemma B.3

$$\text{Term 1} \leq 2\sum_{t=1}^{T}\eta_t\mathbb{E}\left[(1-p_{t,k^*})\sum_{i\neq k^*}p_{t,i}^{1/3} + K^{1/3}\sum_{i\neq k^*}p_{t,i}^{2/3}\right] - \frac{1}{2}\sum_{t=1}^{T}\sum_{i\neq k^*}\mathbb{E}\left[p_{t,i}\right]\Delta_i$$

$$= \sum_{t=1}^{T}\sum_{i\neq k^*}\mathbb{E}\left[\frac{4K^{-1/6}}{\sqrt{t}}(1-p_{t,k^*})p_{t,i}^{1/3} - \frac{1}{4}p_{t,i}\Delta_i\right] + \sum_{t=1}^{T}\sum_{i\neq k^*}\mathbb{E}\left[4\frac{K^{1/6}}{\sqrt{t}}p_{t,i}^{2/3} - \frac{1}{4}p_{t,i}\Delta_i\right]$$

$$= \underbrace{\sum_{t=1}^{T}\sum_{i\neq k^*}\mathbb{E}\left[\frac{4K^{-1/6}}{\sqrt{t}}(1-p_{t,k^*})p_{t,i}^{1/3} - \frac{1}{4}p_{t,i}\Delta_i\right]}_{\text{Term 1.1}} + \underbrace{\sum_{t=1}^{T_0}\sum_{i\neq k^*}\mathbb{E}\left[4\frac{K^{1/6}}{\sqrt{t}}p_{t,i}^{2/3} - \frac{1}{4}p_{t,i}\Delta_i\right]}_{\text{Term 1.2}}$$

$$+ \underbrace{\sum_{t=T_0+1}^{T}\sum_{i\neq k^*}\mathbb{E}\left[4\frac{K^{1/6}}{\sqrt{t}}p_{t,i}^{2/3} - \frac{1}{4}p_{t,i}\Delta_i\right]}_{\text{Term 1.3}},$$

where we used the definition of $\eta_t$ from line 1 to line 2.

The Terms 1.2 and 1.3 can be upper bounded using the same derivation as in the previous calculations for Term 1, where we obtained (15) from (14):

$$\text{Term 12} + \text{Term 13} \leq 8\sqrt{KT_0} + \sum_{t=T_0+1}^{T} \sum_{i \neq k^*} \max_{z \geq 0} \left[ \frac{4K^{1/6}}{\sqrt{t}} z^{2/3} - \frac{\Delta_i}{4} z \right]. \tag{16}$$

Let us bound Term 1.1. Let $\bar{T} := \lceil \frac{256K}{\Delta_*^2} \rceil$. We have

$$\text{Term 11} = \sum_{t=1}^{\bar{T}} \sum_{i \neq k^*} \mathbb{E}\left[ \frac{4K^{-1/6}}{\sqrt{t}}(1 - p_{t,k^*})p_{t,i}^{1/3} - \frac{1}{4}p_{t,i}\Delta_i \right] + \sum_{t=\bar{T}+1}^{T} \sum_{i \neq k^*} \mathbb{E}\left[ \frac{4K^{-1/6}}{\sqrt{t}}(1 - p_{t,k^*})p_{t,i}^{1/3} - \frac{1}{4}p_{t,i}\Delta_i \right].$$

Let $t \geq \bar{T}$, recall that $\sum_{i=1}^{K} p_{t,i} = 1$, therefore using Jensen's inequality for the concave function $x \to x^{1/3}$, we have: $\sum_{i \neq k^*} p_{t,i}^{1/3} \leq K^{2/3}(\sum_{i \neq k^*} p_{t,i})^{1/3} = K^{2/3}(1 - p_{t,k^*})^{1/3}$

$$\sum_{i \neq k^*} \mathbb{E}\left[ \frac{4K^{-1/6}}{\sqrt{t}}(1 - p_{t,k^*})p_{t,i}^{1/3} - \frac{1}{4}p_{t,i}\Delta_i \right] \leq 4\frac{K^{-1/6}}{\sqrt{t}}(1 - p_{t,k^*})K^{2/3}(1 - p_{t,k^*})^{1/3} - \frac{1}{4}\Delta_*(1 - p_{t,k^*})$$

$$\leq \frac{\Delta_*}{4}(1 - p_{t,k^*})^{4/3} - \frac{1}{4}\Delta_*(1 - p_{t,k^*})$$

$$\leq 0,$$

where the the second line follows from the fact that $t \geq \bar{T}$, and the last line from the fact that $p_{t,k^*} \in [0, 1]$. We conclude that

$$\text{Term 1.1} \leq \sum_{t=1}^{\bar{T}} \sum_{i \neq k^*} \mathbb{E}\left[ \frac{4K^{-1/6}}{\sqrt{t}}(1 - p_{t,k^*})p_{t,i}^{1/3} - \frac{1}{4}p_{t,i}\Delta_i \right]$$

$$\leq \sum_{t=1}^{\bar{T}} \sum_{i \neq k^*} \mathbb{E}\left[ \frac{4K^{-1/6}}{\sqrt{t}}p_{t,i}^{1/3} - \frac{1}{4}p_{t,i}\Delta_i \right]$$

$$\leq \sum_{t=1}^{\bar{T}} \sum_{i \neq k^*} \max_{z \geq 0} \left\{ 4\frac{K^{-1/6}}{\sqrt{t}}z^{1/3} - \frac{\Delta_i}{4}z \right\}$$

$$\leq 25 \sum_{t=1}^{\bar{T}} \sum_{i \neq k^*} \frac{K^{-1/4}}{t^{3/4}} \frac{1}{\sqrt{\Delta_i}},$$

where we used Lemma F.4 to obtain the last line. Using $\sum_{t=1}^{\bar{T}} t^{-3/4} \leq 4\bar{T}^{1/4}$, we have

$$\text{Term 1.1} \leq 100 \left( \frac{\bar{T}}{K} \right)^{1/4} \sum_{i \neq k^*} \frac{1}{\sqrt{\Delta_i}}.$$

Next we use the definition of $\bar{T}$, then Jensen's inequality for the concave function $x \to \sqrt{x}$ to have:

$$\text{Term 1.1} \leq 400 \frac{1}{\sqrt{\Delta_*}} \sum_{i \neq k^*} \frac{1}{\sqrt{\Delta_i}}$$

$$\leq 400 \sqrt{\frac{K}{\Delta_*}} \sqrt{\sum_{i \neq k^*} \frac{1}{\Delta_i}}, \tag{17}$$

We conclude combining (16) and (17) that:

$$\text{Term 1} \leq 400 \sqrt{\frac{K}{\Delta_*}} \sqrt{\sum_{i \neq k^*} \frac{1}{\Delta_i}} + 8\sqrt{KT_0} + \sum_{t=T_0+1}^{T} \sum_{i \neq k^*} \max_{z \geq 0} \frac{4K^{1/6}}{\sqrt{t}}z^{2/3} - \frac{\Delta_i}{4}z. \tag{18}$$

**Conclusion:** Combining the bounds in (18) and (15) we have:

$$\mathcal{R}_T \le 400\sqrt{\frac{K}{\Delta_*}}\sqrt{\sum_{i\neq k^*}\frac{1}{\Delta_i}} + 17\sqrt{KT_0} + \sum_{t=T_0+1}^{T}\sum_{i\neq k^*}\max_{z\ge 0}\left\{\frac{7K^{1/6}}{\sqrt{t}}z^{2/3} - \frac{3\Delta_i}{4}z\right\}. \quad (19)$$

Using Lemma F.4, we have:

$$\sum_{t=T_0+1}^{T}\max_{z\ge 0}\left\{\frac{7K^{1/6}}{\sqrt{t}}z^{2/3} - \frac{3\Delta_i}{4}z\right\} \le \sum_{t=T_0+1}^{T}\left(\frac{7K^{1/6}}{\sqrt{t}}\right)^3\left(\frac{3\Delta_i}{4}\right)^{-2}$$

$$\le 610\sum_{t=T_0+1}^{T}\frac{\sqrt{K}}{t^{3/2}}\frac{1}{\Delta_i^2}.$$

We conclude that

$$\sum_{t=T_0+1}^{T}\sum_{i\neq k^*}\max_{z\ge 0}\left\{\frac{7K^{1/6}}{\sqrt{t}}z^{2/3} - \frac{3\Delta_i}{4}z\right\} \le 610\sqrt{K}\sum_{i\neq k^*}S_i(T), \quad (20)$$

where $S_i(T) := \sum_{t=T_0+1}^{T}\frac{1}{\Delta_i^2 t^{3/2}}$. Let us bound the quantities $S_i(T)$. We have for any $i\neq k^*$:

$$S_i(T) \le \sum_{t=T_0+1}^{+\infty}\frac{1}{\Delta_i^2 t^{3/2}}$$

$$\le \frac{1}{\Delta_i^2}\int_{T_0}^{+\infty}\frac{1}{t^{3/2}}dt$$

$$= \frac{1}{\Delta_i^2}\lim_{T\to\infty}\frac{T^{-1/2} - T_0^{-1/2}}{-\frac{1}{2}}$$

$$= \frac{2}{\Delta_i^2\sqrt{T_0}}.$$

Next, we re-inject the bound above on inequality (20) and obtain

$$\sum_{t=T_0+1}^{T}\sum_{i\neq k^*}\max_{z\ge 0}\left\{\frac{7K^{1/6}}{\sqrt{t}}z^{2/3} - \frac{3\Delta_i}{4}z\right\} \le 610\sqrt{K}\sum_{i\neq k^*}S_i(T)$$

$$\le 1220\sqrt{K}\sum_{i\neq k^*}\frac{1}{\Delta_i^2\sqrt{T_0}}.$$

We take

$$T_0 := \lceil * \rceil\frac{1}{\Delta_*}\sum_{i\neq k^*}\frac{1}{\Delta_i}. \quad (21)$$

Therefore:

$$\sum_{t=T_0+1}^{T}\sum_{i\neq k^*}\max_{z\ge 0}\left\{\frac{7K^{1/6}}{\sqrt{t}}z^{2/3} - \frac{3\Delta_i}{4}z\right\} \le 1220\sqrt{K}\sum_{i\neq k^*}\frac{1}{\Delta_i^2\sqrt{T_0}}$$

$$\le 1220\frac{\sqrt{K}}{\sqrt{T_0}}T_0$$

$$\le 1220\sqrt{\frac{K}{\Delta_*}}\sqrt{\sum_{i\neq k^*}\frac{1}{\Delta_i}}. \quad (22)$$

Finally, we use the bounds (21) and (22) in (19) and conclude that for some numerical constant $c$ we have:

$$\mathcal{R}_T \le c\sqrt{\frac{K}{\Delta_*}}\sqrt{\sum_{i\neq k^*}\frac{1}{\Delta_i}}.$$

$\square$

## B.3   Some Lemmas:

The following lemma is an adaptation of part 2 of Lemma 6 in 13 that bounds the stability term.

**Lemma B.3.** *We have*

$$\mathbb{E}\left[\ell_{t,A_t} + \Phi_t(-\hat{L}_t) - \Phi_t(-\hat{L}_{t-1})\right] \leq 2\eta_t \mathbb{E}\left[(1 - p_{t,k^*})\sum_{i \neq k^*} p_{t,i}^{1/3} + K^{1/3}\sum_{i \neq k^*} p_{t,i}^{2/3}\right]$$

*Proof.* We adapt the arguments presented in the proof of Lemma 6 in [13]. We use Lemma B.5, where we choose $x = \mathbb{1}\{B_t = k^*\}\bar{\ell}_{t,k^*}$. When $B_t \neq k^*$, we have $x = 0$ and the expression is maximized for $\tilde{p}_i = p_{t,i}$, since the losses are non-negative, and $\nabla\Psi_t^*$ is monotonically increasing. When $B_t = k^*$, we have $\hat{\ell}_{t,k^*} - x = 0 - x \geq -1$ and we can apply Lemma B.6 and bound $\tilde{p}_i^{4/3}$ by $(3/2)p_{t,i}^{4/3}$. We conclude that

$$\mathbb{E}\left[\sum_{i=1}^{K}\max_{\tilde{p}_i \in [p_{t,i}, \nabla\Psi^*(\nabla\Psi_t(p_t) - \bar{\ell}_t + x\mathbb{1}_K)_i]}\frac{\eta_t}{2}\left(\hat{\ell}_{t,i} - x\right)^2 \left(\tilde{p}_{t,i}\right)^{4/3}\right]$$

$$\leq \sum_{i \neq k^*}\frac{\eta_t}{2}\mathbb{E}\left[\hat{\ell}_{t,i}^2 p_{t,i}^{4/3}\right] + \frac{\eta_t}{2}\mathbb{E}\left[\left(\hat{\ell}_{t,k^*} - x\right)^2 (3/2)^{4/3} p_{t,i}^{4/3}\right] \qquad (23)$$

Let us bound each term in the expression above. For the first term we have:

$$\sum_{i \neq k^*}\frac{\eta_t}{2}\mathbb{E}\left[\hat{\ell}_{t,i}^2 p_{t,i}^{4/3}\right] \leq \sum_{i \neq k^*}\frac{\eta_t}{2}\mathbb{E}\left[\left(\frac{\mathbb{1}(B_t = i \text{ and } A_t \neq B_t')}{p_{t,i}^2} + \frac{\mathbb{1}(B_t = i \text{ and } A_t = B_t')}{r_{t,i}^2}\right)\bar{\ell}_{t,i}^2 p_{t,i}^{4/3}\right]$$

We have:

$$\mathbb{E}_{t-1}\left[\mathbb{1}(B_t = i \text{ and } A_t \neq B_t')\right] = \mathbb{P}_{t-1}\left(A_t \neq B_t'\right)\mathbb{P}\left(B_t = i \mid A_t \neq B_t'\right)$$
$$= (1 - \|\boldsymbol{p}_t\|^2)\,p_{t,i}.$$

Similarly, we show that

$$\mathbb{E}_{t-1}\left[\mathbb{1}\left(B_t = i \text{ and } A_t = B_t'\right)\right] = \|\boldsymbol{p}_t\|^2\,r_{t,i}.$$

Therefore:

$$\sum_{i \neq k^*}\frac{\eta_t}{2}\mathbb{E}\left[\hat{\ell}_{t,i}^2 p_{t,i}^{4/3}\right] \leq \sum_{i \neq k^*}\frac{\eta_t}{2}\mathbb{E}\left[(1 - \|p_t\|^2)p_{t,i}^{1/3} + \frac{\|p_t\|^2}{r_{t,i}}p_{t,i}^{4/3}\right]. \qquad (24)$$

For the second term in (23) we have:

$$\frac{\eta_t}{2}\mathbb{E}\left[\left(\hat{\ell}_{t,k^*} - x\right)^2 (3/2)^{4/3} p_{t,i}^{4/3}\right] \leq \eta_t \mathbb{E}\left[\mathbb{1}(B_t = k^*)\left(\frac{\mathbb{1}\left(A_t \neq B_t'\right)}{p_{t,k^*}} + \frac{\mathbb{1}\left(A_t = B_t'\right)}{r_{t,k^*}} - 1\right)^2 \bar{\ell}_{t,k^*}^2 p_{t,k^*}^{4/3}\right]$$

$$= \eta_t \mathbb{E}\left[\left\{(1 - \|\boldsymbol{p}_t\|^2)p_{t,k^*}\left(\frac{1}{p_{t,k^*}} - 1\right)^2 + \|\boldsymbol{p}_t\|^2 r_{t,k^*}\left(\frac{1}{r_{t,k^*}} - 1\right)^2\right\}\bar{\ell}_{t,k^*}^2 p_{t,k^*}^{4/3}\right]$$

$$\leq \eta_t \mathbb{E}\left[(1 - \|\boldsymbol{p}_t\|^2)(1 - p_{t,k^*})^2 p_{t,k^*}^{1/3} + \|\boldsymbol{p}_t\|^2\frac{(1 - r_{t,k^*})^2}{r_{t,k^*}}p_{t,k^*}^{4/3}\right]$$

$$(25)$$

Finally, we inject the bounds (24) and (25) into (23), rearranging the terms we obtain

$$\mathbb{E}\left[\sum_{i=1}^{K}\max_{\tilde{p}_i\in[p_{t,i},\nabla\Psi^*\left(\nabla\Psi_t(p_t)-\tilde{\ell}_t+x1_K\right)_i]}\frac{\eta_t}{2}\left(\hat{\ell}_{t,i}-x\right)^2\left(\tilde{p}_{t,i}\right)^{4/3}\right]$$

$$\leq\underbrace{\eta_t\mathbb{E}\left[(1-\|p_t\|^2)\left((1-p_{t,k^*})^2p_{t,k^*}^{1/3}+\sum_{i\neq k^*}p_{t,i}^{1/3}\right)\right]}_{\text{Term 1}}$$

$$+\underbrace{\eta_t\mathbb{E}\left[\|p_t\|^2\left(\frac{(1-r_{t,k^*})^2}{r_{t,k^*}}p_{t,k^*}^{4/3}+\sum_{i\neq k^*}\frac{p_{t,i}^{4/3}}{r_{t,i}}\right)\right]}_{\text{Term 2}}$$

**Upper bounding Term 1:** We have:

$$\eta_t\mathbb{E}\left[(1-\|p_t\|^2)\left((1-p_{t,k^*})^2p_{t,k^*}^{1/3}+\sum_{i\neq k^*}p_{t,i}^{1/3}\right)\right]\leq\eta_t\mathbb{E}\left[(1-\|p_t\|^2)\left((1-p_{t,k^*})+\sum_{i\neq k^*}p_{t,i}^{1/3}\right)\right]$$

$$=\eta_t\mathbb{E}\left[(1-\|p_t\|^2)\left(\sum_{i\neq k^*}p_{t,i}+\sum_{i\neq k^*}p_{t,i}^{1/3}\right)\right]$$

$$\leq 2\eta_t\mathbb{E}\left[(1-\|p_t\|^2)\sum_{i\neq k^*}p_{t,i}^{1/3}\right]$$

$$\leq 2\eta_t\mathbb{E}\left[(1-p_{t,k^*})\sum_{i\neq k^*}p_{t,i}^{1/3}\right].$$

**Upper bounding Term 2:** We have:

$$\eta_t\mathbb{E}\left[\|p_t\|^2\left(\frac{(1-r_{t,k^*})^2}{r_{t,k^*}}p_{t,k^*}^{4/3}+\sum_{i\neq k^*}\frac{p_{t,i}^{4/3}}{r_{t,i}}\right)\right]\leq\eta_t\mathbb{E}\left[\frac{(1-r_{t,k^*})^2}{r_{t,k^*}}p_{t,k^*}^{4/3}+\sum_{i\neq k^*}\frac{p_{t,i}^{4/3}}{r_{t,i}}\right]$$

$$\leq\eta_t\mathbb{E}\left[\frac{(1-r_{t,k^*})}{r_{t,k^*}}p_{t,k^*}^{4/3}+\sum_{i\neq k^*}\frac{p_{t,i}^{4/3}}{p_{t,i}^{2/3}}\left(\sum_{j=1}^{K}p_{t,j}^{2/3}\right)\right]$$

$$=\eta_t\mathbb{E}\left[\frac{\sum_{i\neq k^*}r_{t,i}}{p_{t,k^*}^{2/3}}\left(\sum_{j=1}^{K}p_{t,j}^{2/3}\right)p_{t,k^*}^{4/3}+\left(\sum_{i\neq k^*}p_{t,i}^{2/3}\right)\left(\sum_{j=1}^{K}p_{t,j}^{2/3}\right)\right]$$

$$\leq\eta_t\mathbb{E}\left[\left(\sum_{i\neq k^*}p_{t,i}^{2/3}\right)p_{t,k^*}^{2/3}+\left(\sum_{i\neq k^*}p_{t,i}^{2/3}\right)\left(\sum_{j=1}^{K}p_{t,j}^{2/3}\right)\right]$$

$$\leq 2\eta_t K^{1/3}\mathbb{E}\left[\sum_{i\neq k^*}p_{t,i}^{2/3}\right],$$

where we used in the last line Jensen's inequality for the concave function $x\to x^{2/3}$.

$$\square$$

The following lemma is a direct consequence of the second part of Lemma 7 in 13 where we take $\alpha = 2/3$ and $\beta = K^{-1/6}$. It provides a bound on the penalty term:

**Lemma B.4** (Part 2 of Lemma 7 in 13). *For $\eta_t = \frac{2K^{-1/6}}{\sqrt{t}}$, the penalty term satisfies:*

$$
\mathbb{E}\left[\sum_{t=1}^{T} \Phi_t(-\hat{L}_{t-1}) - \Phi_t(-\hat{L}_t) - \ell_{t,k^*}\right] \leq \frac{9}{8} K^{1/6} \sum_{k \neq k^*} \sum_{t=1}^{T} \frac{\mathbb{E}\left[p_{t,k}\right]^{2/3}}{\sqrt{t}} + \frac{9}{4}\sqrt{K}.
$$

*Proof.* Our algorithm uses the Tsallis-inf framework introduced by [22] and [13]. We use a learning rate $\eta_t = \frac{2\beta}{\sqrt{t}}$, with $\beta = K^{-1/6}$. Moreover, the loss estimator we use defined by (11) is unbiased. Therefore the statement of Lemma 7 from [13] applies. The expression in the lemma follows by taking $\alpha = 2/3$ and $\beta = K^{-1/6}$. □

**Lemma B.5.** *Lemma 10 in [13] Let $p_t = \nabla\Phi_t\left(-\tilde{L}_{t-1}\right)$ for $\tilde{L}_t = \tilde{L}_{t-1} + \tilde{\ell}_t$, where $\tilde{\ell}_t$ is an unbiased estimate of $\ell_t$. For any $x \geq 0$, the instantaneous stability of the pseudo-regret of Algorithm 4 satisfies:*

$$
\mathbb{E}\left[\ell_{t,A_t} + \Phi_t\left(-\tilde{L}_t\right) - \Phi_t(-\tilde{L}_{t-1})\right] \leq \mathbb{E}\left[\sum_{i=1}^{K} \max_{\tilde{p}_i \in [p_{t,i}, \nabla\Psi^*\left(\nabla\Psi_t(p_t) - \tilde{\ell}_t + x\mathbf{1}_K\right)_i]} \frac{\eta_t}{2}\left(\tilde{\ell}_{t,i} - x\right)^2 \left(\tilde{p}_{t,i}\right)^{4/3}\right].
$$

*Proof.* Recall that our loss estimators (11) are unbiased and the played arm $A_t$ is selected following the same rule as in [13]. Therefore the statement of Lemma 10 in [13] applies. □

**Lemma B.6.** *Lemma 11 in [13] Let $p \in \mathcal{S}^{K-1}$ and $\tilde{p} = \nabla\Psi_t^*\left(\nabla\Psi_t(p) - \ell\right)$. If $\eta_t \leq 1/4$ then for all $\ell_i \geq -1$ it holds that $\tilde{p}_i^{4/3} \leq \frac{3}{2} p_i^{4/3}$.*

## C  Proof of Theorem 4.2

Proving Theorem 4.2 amounts to combining the previous results. We have using Lemma A.1:

$$
\mathbb{E}\left[R_{T,\mathcal{A}}^{(w)}\right] \leq \mathbb{E}\left[R_{T,\mathcal{A}'}^{(s)}\right].
$$

Furthermore, using Lemma A.2, we have:

$$
\mathbb{E}\left[R_{T,\mathcal{A}'}^{(s)}\right] \leq \mathbb{E}\left[R'_{-1,T}\right].
$$

Now let us show how the problem of upper-bounding the regret above related to the analysis of the modified OMD with Tsallis-INF regularizer in Algorithm 4 developed in Section B.

The regret $R'_{-1,T}$ is with respect to a learner playing with the following strategy: In each round $t \in [T]$

- The learner samples $I'_t$ from $p_t$.
- The learner plays $I'_t$ and incurs $\ell_{t,I'_t} = X_t(I'_t, J'_t)$ where $J'_t$ is independently sampled from $p_t$.
- The learner samples $J_t$ using:
  - from $p_t$ if an event with probability $1 - \|p_t\|^2$ holds.
  - from $r_t$ if an event with probability $\|p_t\|^2$ holds.
- The learner observes the feedback $X_t(J_t, I_t)$ where $I_t$ is independently sampled using $p_t$.

Comparing the strategy above with the game protocol 3 and Algorithm 4 of Section B, we conclude that: $A_t$ plays the same role as $I'_t$, and $J_t$ plays the same role as $B_t$. Moreover, the losses observed $X_t(k, I_t)$ corresponding to $\bar{\ell}_{t,k}$ are independent but follow the same distribution as the actual losses $X_t(k, J'_t)$ corresponding to $\ell_{t,k}$.

Finally in order to apply Theorem B.2, we need to check whether the regret $R'_{-1,T}$ satisfies the necessary self-bounding condition (condition considered in Section B). The last requirement is a direct consequence of Lemma A.2.

Therefore, the bound in Theorem B.2 applies and gives the result.

# D  Proof of Theorem 4.3

We restate here the theorem:

**Theorem D.1.** *Under the assumption of the existence of a Condorcet winner, the weak regret of Algorithm 2 satisfies:*

$$\mathbb{E}\left[R_T^{(w)}\right] \leq c \log(K/\Delta_*) \sum_{k \neq k^*} \frac{K \Delta_{k^*,k}}{\Delta_{j^*(k),k}^2},$$

*where for each* $k \neq k^*$: $j^*(k) \in \arg\max_j \Delta_{j,k}$, $\Delta_* = \min_{k \neq k^*} \Delta_{k^*,k}$ *and* $c = c' \max\{1, \log\log\log(K \vee 16)\}$ *with* $c'$ *an absolute constant.*

*Proof.* **Notation**: Let $\mathcal{F}_t = \sigma\left((I_1, J_1), X_1(I_1, J_1), \ldots, (I_t, J_t), X_t(I_t, J_t)\right)$. Following Algorithm 2, we run EXP3-IX Algorithm for each arm $k$ in each phase (a phase corresponds to a fixed value of the parameter $B$), during the $n^{th}$ increment of the value $B$ (phase number $n$) we have $B = -2^{n-1}$. When we fix arm $k$ as a left arm and run EXP3-IX to choose the right arm for $t$ rounds, we denote by $S(k, n, t)$ the obtained cumulative loss:

$$S(k, n, t) := \sum_{s=1}^{t} \left(X_s(k, J_s) - \frac{1}{2}\right),$$

More rigorously, the sum above is over rounds between $\tau + 1$ and $\tau + t$, where $\tau$ is a variable independent of the considered cumulative loss. We consider the notation above to simplify the proof.

We define, for each $k \in [K]$ and $n \geq 1$, by $\tau_k^{(n)}$ as the stopping time, with respect to $(\mathcal{F}_t)$, corresponding to the round where $(S(k, n, t))_t$ hits the level $-2^{n-1}\sqrt{t}$ for the first time:

$$\tau_k^{(n)} := \min\left\{t \geq 1 : S(k, n, t) < -2^{n-1}\sqrt{t}\right\}.$$

We call phase, the time interval in Algorithm 2 between two increments of the variable $B$. More formally, phase number $n$ corresponds to the number of rounds $t$ such that $\tau_K^{(n)} < t \leq \tau_K^{(n+1)}$.

For $k \leq K$ and a positive $n$, let $E_k^{(n)}$ denote the following event

$$E_k^{(n)} = \{\tau_{k-1}^{(n)} < +\infty\}.$$

We use the following $\tau_K^{(0)} = 0$, $\tau_0^{(n)} := \tau_K^{(n-1)}$ for each $n \geq 1$. The expected regret incurred at phase $n$ satisfies:

$$\mathbb{E}\left[R_T^{(n)} \mid E_1^{(n)}\right] \leq \sum_{k=1}^{K} \mathbb{P}\left(E_k^{(n)} \mid E_1^{(n)}\right) \mathbb{E}\left[\sum_{t=1}^{\tau_k^{(n)}} \min\left\{\Delta_{k^*,k}, \Delta_{k^*,J_t}\right\} \mid E_k^{(n)}\right]$$

$$\leq \sum_{k=1}^{K} \mathbb{P}\left(E_k^{(n)} \mid E_1^{(n)}\right) \mathbb{E}\left[\sum_{t=1}^{\tau_k^{(n)}} \Delta_{k^*,k} \mid E_k^{(n)}\right]$$

$$= \sum_{k=1}^{K} \Delta_{k^*,k} \mathbb{P}\left(E_k^{(n)} \mid E_1^{(n)}\right) \mathbb{E}\left[\tau_k^{(n)} \mid E_k^{(n)}\right].$$

Therefore the weak regret of Algorithm 2 satisfies:

$$\mathbb{E}\left[R_T\right] \leq \sum_{n=1}^{+\infty} \mathbb{P}(E_1^{(n)}) \mathbb{E}\left[R_T^{(n)} \mid E_1^{(n)}\right]$$

$$\leq \sum_{n=1}^{+\infty} \sum_{k=1}^{K} \mathbb{P}(E_k^{(n)}) \Delta_{k^*,k} \mathbb{E}\left[\tau_k^{(n)} \mid E_k^{(n)}\right].$$

Define by $U_k$ the quantity:

$$U_k := \frac{K \log(K)}{\Delta_{j^*(k),k}^2}.$$

Using Lemma D.2. We have for some numerical constant $c > 0$

$$\mathbb{E}\left[R_T\right] \le c\sum_{n=1}^{+\infty}\sum_{k=1}^{K}\mathbb{P}(E_k^{(n)})\Delta_{k^*,k}\left(U_k + \frac{4^n}{\Delta_{j^*(k),k}^2}\right).$$

We have

$$\mathbb{E}\left[R_T\right] \le c\sum_{n=1}^{+\infty}\sum_{k=1}^{K}\mathbb{P}(E_k^{(n)})\Delta_{k^*,k}\left(U_k + \frac{4^n}{\Delta_{j^*(k),k}^2}\right)$$

$$= c\sum_{n=1}^{+\infty}\sum_{k=1}^{K}\mathbb{P}(E_k^{(n)})\Delta_{k^*,k}U_k + \sum_{n=1}^{+\infty}\sum_{k=1}^{K}\mathbb{P}(E_k^{(n)})\Delta_{k^*,k}\frac{4^n}{\Delta_{j^*(k),k}^2}. \tag{26}$$

Let $N = \log_4(1 \vee \log(1 \vee \log_2(1/\Delta_*)))$, therefore we have $\exp(2^{2N}) \ge \log_2(1/\Delta_*)$. Therefore, using Lemma D.3, we have:

$$\sum_{n=1}^{+\infty}\mathbb{P}\left(E_k^{(n)}\right) \le 1 + \sum_{n=2}^{+\infty}\min\left\{1, \frac{\exp(-2^{2n-2})}{4\log(2)}\left(1 - 8\log\left(\min\{1, 2^{n-2}\Delta_*\}\right)\right)\right\}$$

$$\le 1 + \sum_{n=2}^{+\infty}\min\left\{1, \frac{\exp(-2^{2n-2})}{4\log(2)}9\log\left(1/\Delta_*\right)\right\}$$

$$\le 1 + \sum_{n=2}^{+\infty}\min\left\{1, 5\exp(2^{2N} - 2^{2n-2})\right\}$$

$$\le 1 + N + \sum_{n=(N+2)\vee 2}^{\infty}5\exp(-2^{2(n-N)-2}) \le N + 4. \tag{27}$$

Moreover, we have:

$$\sum_{n=1}^{+\infty}4^n\mathbb{P}\left(E_k^{(n)}\right) \le 4 + \sum_{n=2}^{+\infty}4^n\frac{\exp(-2^{2n-2})}{4\log(2)}\left(1 - 8\log\left(\min\{1, 2^{n-2}\Delta_*\}\right)\right)$$

$$\le 4 + \frac{9\log(1/\Delta_*)}{4\log(2)}\sum_{n=2}^{+\infty}4^n\exp(-2^{n-2})$$

$$\le 4 + 150\log(1/\Delta_*). \tag{28}$$

Finally we plug (27) and (28) into (26) and use the fact that $K\log(K)N + \log(1/\Delta_*) \le 2K\log(K/\Delta_*)\max\{1, \log\log\log(K \vee 16)\}$, to conclude. $\qquad\square$

## D.1 Auxiliary Lemmas

We recall the notation: for each $k \in [K]$, let $j^*(k) \in \arg\min_{j\in[K]\setminus\{k\}}\Delta_{k,j}$. Therefore, $j^*(k)$ represents the arm with the largest chance to beat arm $k$. Moreover, let $S(k, n, t)$ denote the cumulative loss obtained when running EXP3-IX algorithm in phase $n$ for arm $k$. For $n \ge 1$, let $\tau_k^{(n)}$ denote the stopping time corresponding to the round where the process $S(k, n, t)$ hits the level $-2^{n-1}\sqrt{t}$:

$$\tau_k^{(n)} := \min\left\{t \ge 1 : S(k, n, t) \le -2^{n-1}\sqrt{t}\right\}.$$

**Lemma D.2.** *Let* $k \in [K] \setminus \{k^*\}$*, then we have for any* $n > 0$*:*

$$\mathbb{E}\left[\tau_k^{(n)} \mid \tau_{k-1}^{(n)} < +\infty\right] \le c\frac{K\log(K)}{\Delta_{j^*(k),k}^2} + c\frac{4^n}{\Delta_{j^*(k),k}^2},$$

*where $c$ is an absolute constant.*

*Proof.* Fix $k \in [K] \setminus \{k^*\}$ and $n \ge 1$. Suppose that $\{\tau_{k-1}^{(n)} < +\infty\}$ holds. Recall the definition of $S(k, n, t)$:

$$S(k, n, t) := \sum_{s=1}^{t}\left(X_s(k, J_s) - \frac{1}{2}\right).$$

To ease notation we will focus only on the rounds where the fixed arm $k$ was chosen as a left arm in phase $n$. Recall that at each phase when we consider a new arm, we run EXP3-IX from scratch, therefore the obtained cumulative loss process is independent from the past. Denote by $Y_{j,s}$ the sample received when choosing $j$ as a right arm at round $s$, $Y_{j,s}$ has the same distribution as the variable $X(k,j) - 1/2$, hence $\mathbb{E}[Y_{j,u}] = \Delta_{k,j}$. In each round $u$, the chosen arm is denoted $A_u$. Therefore we have:

$$S(k,n,t) = \sum_{s=1}^{t} Y_{A_s,s}.$$

Let $j^* \in \arg\min_j \{\Delta_{k,j}\}$ $(j^* = j^*(k)$, we just dropped the dependence on $k$). Let

$$\Delta := \Delta_{k,j^*} < 0, \tag{29}$$

$\Delta$ is negative because $k$ is not the Condorcet winner (remember we fixed $k \in [K] \setminus \{k^*\}$). Define the (random) regret for this problem after $t$ rounds as follows:

$$R_t = \sum_{s=1}^{t} Y_{A_s,s} - \sum_{s=1}^{t} Y_{j^*,s}.$$

Our objective is to upper-bound the expectation of the stopping time $\tau_k^{(n)}$. To develop such a bound we use the identity:

$$\mathbb{E}\left[\tau_k^{(n)} \mid \tau_{k-1}^{(n)} < +\infty\right] = \sum_{N=0}^{+\infty} \mathbb{P}\left(\tau_k^{(n)} > N \mid \tau_{k-1}^{(n)} < +\infty\right).$$

In the remainder of this proof all events are assumed to hold conditionally to $\{\tau_{k-1}^{(n)} < +\infty\}$. Let us bound the probabilities in the rhs: Fix $m \geq n$ and let

$$N_m := \lceil \frac{K}{\Delta^2} + \frac{2^m}{\Delta^2} \rceil, \tag{30}$$

where $\Delta$ is defined by (29). Recall that conditional to $\{\tau_{k-1}^{(n)} < +\infty\}$ the event $\{\tau_k^{(n)} > N_m\}$ implies in particular that $\{\sum_{t=1}^{N_m} Y_{A_t,t} > -2^{n-1}\sqrt{N_m}\}$. Therefore:

$$\mathbb{P}\left(\tau_k^{(n)} > N_m\right) \leq \mathbb{P}\left(\sum_{t=1}^{N_m} Y_{A_t,t} > -2^{n-1}\sqrt{N_m}\right).$$

Let us upper bound the probability of the last event. We have

$$\mathbb{P}\left(\sum_{t=1}^{N_m} Y_{A_t,t} > -2^{n-1}\sqrt{N_m}\right) = \mathbb{P}\left(\sum_{t=1}^{N_m} Y_{A_t,t} - \sum_{t=1}^{N_m} Y_{j^*,t} + \sum_{t=1}^{N_m} Y_{j^*,t} - N_m\Delta > -2^{n-1}\sqrt{N_m} - N_m\Delta\right)$$

$$= \mathbb{P}\left(R_{N_m} + \sum_{t=1}^{N_m} Y_{j^*,t} - N_m\Delta > -2^{n-1}\sqrt{N_m} - N_m\Delta\right)$$

$$\leq \mathbb{P}\left(R_{N_m} > 7x\sqrt{KN_m \log(K)}\right)$$

$$+ \mathbb{P}\left(\sum_{t=1}^{N_m} Y_{j^*,t} - N_m\Delta > -2^{n-1}\sqrt{N_m} - N_m\Delta - 7x\sqrt{KN_m \log(K)}\right),$$

where $x$ is any constant larger than 1. Using the definition of $N_m$ (recall that $\Delta < 0$), we have $-N_m \Delta \geq -2^m/\Delta$. Therefore

$$
\begin{aligned}
\mathbb{P}\left(\sum_{t=1}^{N_m} Y_{A_t,t} > -2^{n-1}\sqrt{N_m}\right) &\leq \mathbb{P}\left(R_{N_m} > 7x\sqrt{KN_m \log(K)}\right) \\
&\quad + \mathbb{P}\left(\sum_{t=1}^{N_m} Y_{j^*,t} - N_m\Delta > -2^{n-1}\sqrt{N_m} - \frac{2^m}{\Delta} - 7x\sqrt{KN_m \log(K)}\right) \\
&\leq \mathbb{P}\left(R_{N_m} > 7x\sqrt{KN_m \log(K)}\right) \\
&\quad + \mathbb{P}\left(\sum_{t=1}^{N_m} Y_{j^*,t} - N_m\Delta > -2^{n-1}\sqrt{N_m} - \frac{2^m}{\Delta} - 7x\sqrt{KN_m \log(K)}\right).
\end{aligned}
\tag{31}
$$

Now let us take:
$$
x = \frac{-2^m}{28|\Delta|\sqrt{KN_m \log(K)}}.
$$

Define $\bar{n}$ as the smallest integer in $[2n+3, +\infty)$ such that
$$
2^{\bar{n}} \geq 28\Delta\sqrt{KN_{\bar{n}} \log(K)}.
\tag{32}
$$

Hence $2^{\bar{n}} \geq K$. Moreover, for $m \geq \bar{n}$, we have: $x \geq 1$. Furthermore:

$$
\begin{aligned}
2^{n-1}\sqrt{N_m} &< 2^{n-1}\sqrt{4\max\left\{\frac{K}{\Delta^2}, \frac{2^m}{\Delta^2}\right\}} \\
&\leq 2^{n-1}\sqrt{4\frac{2^m}{\Delta^2}} = -\frac{2^{\frac{m}{2}+n}}{\Delta} \\
&\leq -\frac{2^{m-3/2}}{\Delta},
\end{aligned}
\tag{33}
$$

where we used $m \geq \bar{n} \geq 2n+3$.

Using (31):
$$
\mathbb{P}\left(\sum_{t=1}^{N_m} Y_{A_t,t} > -2^{n-1}\sqrt{N_m}\right) \leq \mathbb{P}\left(R_{N_m} > 7x\sqrt{KN_m \log(K)}\right) + \mathbb{P}\left(\sum_{t=1}^{N_m} Y_{j^*,t} - N_m\Delta > -\frac{2^{m-2}}{\Delta}\right).
$$

Using Theorem D.5 to bound the first term (recall that by definition of $\bar{n}$, for $m \geq \bar{n}$: $x \geq 1$), and Hoeffding's inequality to bound the second term we obtain:

$$
\mathbb{P}\left(\sum_{t=1}^{N_m} Y_{A_t,t} > -2^{n-1}\sqrt{N_m}\right) \leq 2\exp\left(-x\sqrt{\log(K)}\right) + \exp\left(-\frac{2^{2m-4}}{\Delta^2 N_m}\right).
\tag{34}
$$

Let us upper-bound the r.h.s of the inequality above. We have for $m \geq \bar{n}$:

$$
\exp\left(-x\sqrt{\log(K)}\right) = \exp\left(-\frac{2^m\sqrt{\log(K)}}{28|\Delta|\sqrt{KN_m \log(K)}}\right)
$$

Using (30) we have that $N_m \leq \frac{2^m}{\Delta^2}$, therefore $\frac{1}{\sqrt{N_m}} \geq \frac{|\Delta|}{2^{m/2}}$ we get

$$
\begin{aligned}
\exp\left(-x\sqrt{\log(K)}\right) &\leq \exp\left(-\frac{2^m\sqrt{\log(K)}}{28\Delta\sqrt{K\log(K)}} \cdot \frac{|\Delta|}{2^{m/2}}\right) \\
&\leq \exp\left(-\frac{2^{m/2}}{28}\sqrt{\frac{1}{K}}\right),
\end{aligned}
\tag{35}
$$

where we used the definition of $N_m$. The second term in (34) can be bounded for $m \geq \bar{n}$ following:

$$\exp\left(-\frac{2^{2m-4}}{\Delta^2 N_m}\right) \leq \exp\left(-2^{m-4}\right). \tag{36}$$

Combining (34), (35) and (36) we conclude that:

$$\mathbb{P}\left(\sum_{t=1}^{N_m} Y_{A_t,t} > -2^{n-1} - \sqrt{N_m}\right) \leq 2\exp\left(-\frac{2^{m/2}}{28}\sqrt{\frac{1}{K}}\right) + \exp\left(-2^{m-4}\right)$$

$$\leq 3\exp\left(-\frac{2^{m/2}}{28}\sqrt{\frac{1}{K}}\right). \tag{37}$$

We conclude that for $m \geq \bar{n}$:

$$\mathbb{P}\left(\tau_k^{(n)} > N_m \mid \tau_{k-1}^{(n)} < +\infty\right) \leq 3\exp\left(-\frac{2^{m/2}}{28}\sqrt{\frac{1}{K}}\right).$$

We have:

$$\mathbb{E}\left[\tau_k^{(n)} \mid \tau_{k-1}^{(n)} < +\infty\right] = \sum_{N=0}^{+\infty} \mathbb{P}\left(\tau_k^{(n)} > N \mid \tau_{k-1}^{(n)} < +\infty\right)$$

$$\leq N_{\bar{n}} + \sum_{N=N_{\bar{n}}}^{+\infty} \mathbb{P}\left(\tau_k^{(n)} > N \mid \tau_{k-1}^{(n)} < +\infty\right)$$

$$\leq N_{\bar{n}} + \sum_{m=\bar{n}}^{+\infty}\sum_{N=N_m}^{N_{m+1}-1} \mathbb{P}\left(\tau_k^{(n)} > N \mid \tau_{k-1}^{(n)} < +\infty\right)$$

$$\leq N_{\bar{n}} + \sum_{m=\bar{n}}^{+\infty}(N_{m+1} - N_m)\mathbb{P}\left(\tau_k^{(n)} > N_m \mid \tau_{k-1}^{(n)} < +\infty\right)$$

$$\leq N_{\bar{n}} + 3\sum_{m=\bar{n}}^{+\infty}\left(\frac{2^m}{\Delta^2} + 1\right)\exp\left(-\frac{2^{m/2}}{28}\sqrt{\frac{1}{K}}\right)$$

$$= N_{\bar{n}} + 6\frac{2^{\bar{n}}}{\Delta^2}\sum_{m=\bar{n}}^{+\infty} 2^{m-\bar{n}}\exp\left(-\frac{2^{(m-\bar{n})/2}}{28}\sqrt{\frac{2^{\bar{n}}}{K}}\right).$$

Recall that by definition of $\bar{n}$ in (32), we have: $2^{\bar{n}} \geq K$. Therefore:

$$\mathbb{E}\left[\tau_k^{(n)} \mid \tau_{k-1}^{(n)} < +\infty\right] \leq N_{\bar{n}} + 6\frac{2^{\bar{n}}}{\Delta^2}\sum_{m=\bar{n}}^{+\infty} 2^{m-\bar{n}}\exp\left(-\frac{2^{(m-\bar{n})/2}}{28}\right)$$

$$\leq N_{\bar{n}} + 6\frac{2^{\bar{n}}}{\Delta^2}\sum_{p=0}^{+\infty} 2^p\exp\left(-\frac{2^{p/2}}{28}\right).$$

Next we use the bound

$$\sum_{p=0}^{+\infty} 2^p\exp\left(-\frac{2^{p/2}}{28}\right) \leq 888.$$

We conclude using the definition of $N_m$ and $\bar{n}$ that

$$\mathbb{E}\left[\tau_k^{(n)} \mid \tau_{k-1}^{(n)} < +\infty\right] \leq \frac{K}{\Delta^2} + \frac{2^{\bar{n}}}{\Delta^2} + 6000\frac{2^{\bar{n}}}{\Delta^2}$$

$$\leq \frac{K}{\Delta^2} + c\max\left\{\frac{2^{2n}}{\Delta^2}, \frac{K\log(K)}{\Delta^2}\right\}$$

$$\leq (c+1)\frac{K\log(K)}{\Delta^2} + c\frac{4^n}{\Delta^2},$$

where $c$ is an absolute constant. $\qquad\square$

**Lemma D.3.** *We have for any $n \geq 2$, $k \in [K]$:*

$$\mathbb{P}\left(E_k^{(n)}\right) \leq \frac{\exp(-2^{2n-2})}{4\log(2)}\left(1 - 8\log(\min\{1, 2^{n-2}\Delta_*\})\right),$$

*where $\Delta_* = \min_{k \neq k^*} \Delta_{k^*,k}$.*

*Proof.* Let $n \geq 2$ and $k \in [K]$. We have that $E_k^{(n)}$ implies the event: $E_{k^*+1}^{(n-1)}$. Therefore:

$$\begin{aligned}
\mathbb{P}\left(E_k^{(n)}\right) &\leq \mathbb{P}\left(E_{k^*+1}^{(n-1)}\right) \\
&= \mathbb{P}\left(\tau_{k^*}^{(n-1)} < +\infty\right) \\
&\leq \frac{\exp(-2^{2n-2})}{4\log(2)}\left(1 - 8\log(\min\{1, 2^{n-2}\Delta_*\})\right),
\end{aligned}$$

where we used in the last line Lemma F.3. $\qquad\square$

## D.2 Deviation guarantees for the regret of EXP3-IX

We recall below the algorithm EXP3-IX from [11]

---
**Algorithm 5** EXP3-IX
---
**Input**: $(\eta_t)_t$, $(\gamma_t)_t$.
**Initialization**: $w_{i,1} = 1$ for all $i$.
**for** $t = 1, \ldots$ **do**
  Let $p_{t,i} = \frac{w_{i,t}}{\sum_{j=1}^K w_{j,t}}$.
  Draw $A_t \sim \mathbf{p}_t = (p_{1,t}, \ldots, p_{K,t})$.
  Observe the loss $\ell_{A_t,t}$.
  $\tilde{\ell}_{i,t} \leftarrow \frac{\ell_{i,t}}{p_{i,t}+\gamma}\mathbb{1}(I_t = i)$ for all $i \in [K]$.
  $w_{t+1,i} \leftarrow w_{t,i}\exp(-\eta_t\tilde{\ell}_{t,i})$ for all $i \in [K]$.
**end for**

---

Define the random regret $R_T$ by: $R_T := \sum_{t=1}^T \ell_{I_t,t} - \min_{i \in [K]} \sum_{t=1}^T \ell_{i,t}$. We restate below the main result from [11].

**Theorem D.4.** *Theorem 1 in [11]. Fix an arbitrary $\delta \in (0,1)$, set $\eta_t = 2\gamma_t = \sqrt{\frac{\log(K)}{Kt}}$ for all t, then EXP3-IX guarantees with probability at least $1 - \delta$:*

$$R_T \leq 4\sqrt{KT\log(K)} + \left(2\sqrt{\frac{KT}{\log(K)}} + 1\right)\log(2/\delta).$$

We have the following corollary

**Corollary D.5.** *Fix $x \geq 1$, and consider EXP3-IX algorithm with $\eta_t = 2\gamma_t = \sqrt{\frac{\log(K)}{Kt}}$, then:*

$$\mathbb{P}\left(R_T \leq 7x\sqrt{KT\log(K)}\right) \leq 2\exp\left(-x\sqrt{\log(K)}\right).$$

*Proof.* Let $x \geq 1$, take:

$$\delta = \min\left\{1, 2\exp\left(-x\sqrt{\log(K)}\right)\right\}.$$

Therefore:

$$x \geq \frac{\log(2/\delta)}{\sqrt{\log(K)}}.$$

We have with probability at least $1 - \delta \geq 1 - 2\exp(-x\sqrt{\log(K)})$:

$$R_T \leq 4\sqrt{KT\log(K)} + \left(2\sqrt{\frac{KT}{\log(K)}} + 1\right)\log(2/\delta)$$

$$\leq 7\sqrt{KT}\max\left\{\sqrt{\log(K)}, \log(2/\delta)\right\}$$

$$= 7\sqrt{KT\log(K)}\max\left\{1, \frac{\log(2/\delta)}{\sqrt{\log(K)}}\right\}$$

$$\leq 7x\sqrt{KT\log(K)}.$$

$\square$

## E  Proof of Theorem 5.1

We restate the theorem for the lower bound, then we proceed with the proof. Let $\Delta_{\mathrm{cw}}$ denote a positive number. For a dueling bandits problem, we denote by $M = (M_{i,j})_{1 \geq i,j \leq K}$ the matrix such that $M_{i,j} = \Delta_{i,j}$. Define the class of problems $\mathbb{D}(\Delta_{\mathrm{cw}})$ by the set of matrices $M$ representing the gaps $(\Delta_{i,j})_{ij}$ such as $M$ is skew-symmetric and there exists some $k^* \in [K]$ such that

$$\begin{cases} \forall i \neq k^* : M_{k^*,i} = \Delta_{\mathrm{cw}} \\ \text{and} \\ \forall i,j \neq k^* : |M_{i,j}| \leq \Delta_{\mathrm{cw}} \end{cases}$$

where $k^* \in [K]$ denotes the Condorcet winner.

**Theorem E.1.** *Fix $K \geq 6$, $\Delta_{cw} \in (0, 1/4)$. The weak regret of an algorithm $\mathcal{A}$ satisfies:*

$$\max_{M \in \mathbb{D}(\Delta_{cw})} \mathbb{E}_{M,\mathcal{A}}[R_T] \geq c\frac{K}{\Delta_{cw}},$$

*when $T \geq c'K/\Delta_{cw}^2$. Here $c$ and $c'$ are numerical constants.*

*Proof.* Let $M^{(0)}$ denote a matrix such that $M_{1,i}^{(0)} = \Delta_{cw}$ for any $i > 1$ and $M_{i,j}^{(0)} = 0$ for $i, j \neq 1$. Let $\mathbb{P}_0$ denote the probability distribution associated with the dueling bandits' problem with matrix $M^{(0)}$ (where arm 1 is the Condorcet winner).

Let $k \neq 1$, let $M^{(k)}$ denote the matrix defined by: for all $u \neq k$ $M_{k,u}^{(k)} = \Delta_{cw} = -M_{u,k}^{(k)}$, for all $i \neq k$ $M_{1,i}^{(k)} = \Delta_{\mathrm{cw}} = -M_{i,1}^{(k)}$, otherwise for $i \neq k, 1$ and $j \neq k, 1$: $M_{i,j}^{(k)} = 0$. Let $\mathbb{P}_k$ denote the probability distribution associated with the dueling bandits' problem with matrix $M^{(k)}$ (where arm $k$ is the Condorcet winner).

For any $u \in [K]$, let $N_u$ denote the total number of rounds where arm $u$ was queried. For $u, v \in [K]$, let $N_{u,v}$ denote the total number of rounds where arms $u$ and $v$ were dueled.

**Information theoretic tool:**  Without loss of generality, we assume that the player follows a deterministic strategy $\mathcal{A}$. Let us introduce the following notation: let $Z_t = \big((I_t, J_t), X_t(I_t, J_t)\big)$ denote the information disclosed to the player at time $t$. Let $\boldsymbol{Z}_t = (Z_1, \ldots, Z_t)$ denote the entire information available to the player after $t$ rounds.

**Lemma E.2.** *Assume that $\Delta_{cw} \leq 1/4$. Let $F(\boldsymbol{Z}_T)$ denote a fixed function of the player observations, taking values in $[0, B]$. Then for any $k \in [K] \setminus \{1\}$ and any player strategy $\mathcal{A}$,*

$$\mathbb{E}_k[F(\boldsymbol{Z}_T)] \leq \mathbb{E}_0[F(\boldsymbol{Z}_T)] + 4B\sqrt{\frac{2}{3}\Delta_{cw}^2\mathbb{E}_0[N_k]},$$

*where $N_u = \sum_{v=1}^K N_{u,v}$ and $N_{u,v}$ denotes the number of rounds where arms $u$ and $v$ were duelled.*

*Proof.* Recall that for any function $G$ bounded by $R$, we have: $|\mathbb{E}_{X \sim \mathbb{P}}[G(X)] - \mathbb{E}_{X \sim \mathbb{Q}}[G(X)]| \leq 2R\,\mathrm{TV}(\mathbb{P}, \mathbb{Q})$, where $\mathrm{TV}(.,.)$ denotes the total variation distance. Therefore, by shifting $F$ by $-B/2$, we have:

$$\mathbb{E}_k F(\boldsymbol{Z}_T) - \mathbb{E}_0 F(\boldsymbol{Z}_T) \leq B\,\mathrm{TV}(\mathbb{P}_k, \mathbb{P}_0) \leq B\sqrt{\frac{1}{2}\mathrm{KL}\left(\mathbb{P}_0, \mathbb{P}_k\right)},$$

by Pinsker's inequality, where $\mathrm{KL}(.)$ denotes the Kullback-Leibler divergence and KL(x,y) for $x, y \in (0,1)$ denotes the Kullback-Leibler divergence between two Bernoulli distributions with means $x$ and $y$.

Next we use the chain rule for relative entropy (Theorem 2.5.3 in 5):

$$\mathrm{KL}\left(\mathbb{P}_0, \mathbb{P}_k\right) = \sum_{t=1}^{T} \mathbb{E}\left[\mathrm{KL}\left(\mathbb{P}_0\left(Z_t | \boldsymbol{Z}_{t-1}\right), \mathbb{P}_k\left(Z_t | \boldsymbol{Z}_{t-1}\right)\right)\right]$$

Observe that we have for each $t \in [T]$:

$$
\begin{aligned}
&\mathbb{E}\left[\mathrm{KL}\left(\mathbb{P}_0\left(Z_t | \boldsymbol{Z}_{t-1}\right), \mathbb{P}_k\left(Z_t | \boldsymbol{Z}_{t-1}\right)\right)\right] \\
&\leq \quad \sum_{u \neq k} \mathbb{P}_0\left(k, u \in \{I_t, J_t\}\right) \mathrm{KL}(M_{k,u}^{(0)} + 1/2; M_{k,u}^{(k)} + 1/2) \\
&\leq \quad \frac{64}{3}\mathbb{P}_0\left(k \in \{I_t, J_t\}\right) \Delta_{cw}^2,
\end{aligned}
$$

where in the last line we used that $KL(x, y) \leq (x - y)^2 / [y(1 - y)]$. Summing over $t \in [T]$ leads to the desired result. $\qquad\square$

Recall that the weak regret for the problem $\mathbb{P}_k$ is given by:

$$
\begin{aligned}
\mathbb{E}_k\left[R_T\right] &= \sum_{u,v=1}^{K} \min\{\Delta_{k,u}, \Delta_{k,v}\}\mathbb{E}_k[N_{u,v}] \\
&= \Delta_{cw}(T - \mathbb{E}_k[N_k]).
\end{aligned}
$$

Applying Lemma E.2 with $F(Z_T) = \Delta_{cw} N_k$, we have:

$$\Delta_{cw}\mathbb{E}_0\left[T - N_k\right] \leq \Delta_{cw}\mathbb{E}_k[T - N_k] + 2\Delta_{cw}T\sqrt{\frac{8}{3}\Delta_{cw}^2\mathbb{E}_0[N_k]}$$

Averaging over $k \in [K] \setminus \{1\}$ and using Jensen's inequality:

$$\Delta_{\mathrm{cw}}\mathbb{E}_0\left[T - \frac{1}{K-1}\sum_k N_k\right] \leq \frac{1}{K-1}\sum_k \mathbb{E}_k\left[R_T\right] + 2\Delta_{cw}T\sqrt{\Delta_{cw}^2 \frac{8}{2(K-1)}\mathbb{E}_0[\sum_k N_k]},$$

Observe that $\sum_{u=1}^{K} N_u \leq 2T$, therefore the inequality above gives:

$$\Delta_{\mathrm{cw}}\left(T - \frac{2T}{K-1}\right) \leq \frac{1}{K-1}\sum_k \mathbb{E}_k\left[R_T\right] + 2\Delta_{cw}T\sqrt{\Delta_{cw}^2 \frac{16T}{3(K-1)}}.$$

Let $\mathbb{P}_*$ denote the problem where we choose $k$ uniformly at random from $[K] \setminus \{1\}$, then we proceed to the game where gaps are given by $M^{(k)}$. Hence we have:

$$\mathbb{E}_*\left[R_T\right] = \frac{\sum_k \mathbb{E}_k\left[R_T\right]}{K-1}.$$

We conclude that

$$\mathbb{E}_*[R_T] \geq \Delta_{\mathrm{cw}}\left(T - \frac{2T}{K-1}\right) - 2\Delta_{cw}T\sqrt{\frac{16T\Delta_{cw}^2}{3K-3}}.$$

Therefore,

$$\mathbb{E}_*[R_T] \geq \sup_{T' \leq T} \mathbb{E}_* [R_{T'}]$$

$$\geq \sup_{T' \leq T} \left\{ \Delta_{\mathrm{cw}} \left( T' - \frac{2T'}{K-1} \right) - 2\Delta_{cw} T' \sqrt{\frac{16T'\Delta_{cw}^2}{3(K-1)}} \right\}. \tag{38}$$

Recall that we assume that $K \geq 6$. In the above inequality, the supremum is achieved for $T'$ of the order of $K/\Delta_{cw}^2$, which is achievable as long as $T$ is at least of this order. This leads to the desired result when $K/\Delta_{cw}^2$ is higher than some numerical constant $c''$. In the extreme situation, where $K/\Delta_{cw}^2$ is smaller or equal to $c''$, the lower bound (38) could be negative for any $T' \geq 1$. In that case, we can use the trivial bound $\mathbb{E}_*[R_T] \geq \mathbb{E}_*[R_1] \geq c'\Delta_{cw}$ as, with probability bounded away from zero, the first duel does not contain the Condorcet winner.

$\square$

## F   Technical Results:

**Lemma F.1.** *Let* $q \in (0,1)$*, we have:*

$$\sum_{n=0}^{+\infty} 2^n q^{2^n} \leq 2 \sum_{n=1}^{+\infty} q^n$$

*Proof.* We have:

$$\sum_{n=1}^{+\infty} 2^{n-1} q^{2^n} \leq \sum_{n=1}^{+\infty} \sum_{i=0}^{2^{n-1}} q^{2^{n-1}+i}$$

$$\leq \sum_{n=1}^{\infty} q^n.$$

$\square$

**Lemma F.2** (Doob's maximal inequality: Section 5.6 from Lawler)**.** *Let* $(X_i)$ *be a sequence of independent Bernoulli variables such that for any* $i \geq 1$*:* $\mathbb{E}[X_i] = p_i$*. Let* $S_t = \sum_{i=1}^{t}(X_i - p_i)$*. We have for any* $t \geq 1$*,* $a > 0$*:*

$$\mathbb{P}\left( \max_{1 \leq t \leq n} \{S_t\} > a \right) \leq \exp\left( -\frac{2a^2}{n} \right).$$

*Proof.* We have $S_t$ is a martingale with respect to the filtration associated to the process $(X_i)$. Therefore using Doob's maximal inequality: Section 5.6 [9], for any $b > 0$:

$$\mathbb{P}\left( \max_{1 \leq t \leq n} \{S_t\} > a \right) \leq \frac{\mathbb{E}\left[ \exp(bS_n) \right]}{\exp(ba)}$$

$$= \frac{\mathbb{E}\left[ \exp(b \sum_{i=1}^{n}(X_i - p_i)) \right]}{\exp(ba)}$$

$$\leq \exp(-ba) \exp(nb^2/8),$$

where we used the fact that for each $i \in [n] : \mathbb{E}[\exp(b(X_i - \mathbb{E}[X_i]))] \leq \exp(b^2/8)$. The conclusion follows by minimizing the upper bound for $b > 0$.

$\square$

**Lemma F.3.** *Let* $(X_t)$ *be a sequence of independent Bernoulli variables such that for some* $\Delta \in (0, 1/2)$*, for each* $t$ *we have:* $\mathbb{E}[X_t] \geq \frac{1}{2} + \Delta$*. Let* $B \geq 1$ *and define the stopping time:*

$$\tau := \inf \left\{ t \geq 1 : \sum_{s=1}^{t} \left( X_s - \frac{1}{2} \right) < -B\sqrt{t} \right\}.$$

*Then we have:*

$$\mathbb{P}\left(\tau < +\infty\right) \leq \frac{\exp(-B^2)}{4\log(2)}\left(1 - 8\log(\min\{1, B\Delta/2\})\right).$$

*Proof.* Let $S_t = \sum_{i=1}^{t}(\mathbb{E}[X_i] - X_i)$. We have

$$\mathbb{P}(\tau < +\infty) = \mathbb{P}\left(\exists t \in \mathbb{N} : \sum_{s=1}^{t}\left(X_s - \frac{1}{2}\right) < -B\sqrt{t}\right)$$

$$= \mathbb{P}\left(\exists t \in \mathbb{N} : \sum_{s=1}^{t}\left(\frac{1}{2} + \Delta - X_s\right) > \Delta t + B\sqrt{t}\right)$$

$$\leq \mathbb{P}\left(\exists t \in \mathbb{N} : \sum_{s=1}^{t}\left(\mathbb{E}[X_s] - X_s\right) > \Delta t + B\sqrt{t}\right)$$

$$= \mathbb{P}\left(\exists t \in \mathbb{N} : S_t > \Delta t + B\sqrt{t}\right)$$

Since we have for each $t \geq 1$: $S_t \leq t$, and $\Delta > 0$. We have:

$$\mathbb{P}\left(\tau < +\infty\right) = \mathbb{P}\left(\exists t \geq 1 : S_t > \Delta t + B\sqrt{t}\right)$$

$$= \mathbb{P}\left(\exists t \geq B^2 : S_t > \Delta t + B\sqrt{t}\right).$$

Let $N_0 = \lfloor \log_2(B^2) \rfloor$. We have

$$\mathbb{P}\left(\tau < +\infty\right) \leq \sum_{n=N_0}^{+\infty} \mathbb{P}\left(\exists t \in [2^n, 2^{n+1}] : S_t > \Delta t + B\sqrt{t}\right)$$

$$\leq \sum_{n=N_0}^{+\infty} \mathbb{P}\left(\exists t \in [2^n, 2^{n+1}] : S_t > \Delta 2^n + B2^{n/2}\right)$$

$$\leq \sum_{n=N_0}^{+\infty} \mathbb{P}\left(\exists t \leq 2^{n+1} : S_t > \Delta 2^n + B2^{n/2}\right).$$

Using Lemma F.2, we have:

$$\mathbb{P}\left(\exists t \leq 2^{n+1} : S_t > \Delta 2^n + 2B2^{n/2}\right) = \mathbb{P}\left(\max_{1 \leq t \leq 2^{n+1}} S_t > \Delta 2^n + B2^{n/2}\right)$$

$$\leq \exp\left(-\frac{\left(\Delta 2^n + B2^{n/2}\right)^2}{2^n}\right)$$

$$\leq \exp\left(-\Delta^2 2^n - B^2\right).$$

Pluging the last bound in the previous display gives:

$$\mathbb{P}\left(\tau < +\infty\right) \leq \sum_{n=N_0}^{+\infty} \exp\left(-\Delta^2 2^n - B^2\right)$$

$$\leq \exp(-B^2) \sum_{n=1}^{+\infty} \exp\left(-\frac{\Delta^2}{2} 2^{n+N_0}\right)$$

$$\leq \exp(-B^2) \sum_{n=1}^{+\infty} \exp\left(-\frac{1}{4}(B\Delta)^2 2^n\right)$$

$$\leq \exp(-B^2)\left(\frac{1}{4\log(2)} - \frac{2\log(\min\{1, B\Delta/2\})}{\log(2)}\right).$$

To get the last bound, we bounded the sum by an integral and use the change of variable $u = (B\Delta)^2 2^t/4$.

$$
\sum_{n=1}^{+\infty} \exp\left(-\frac{1}{4}(B\Delta)^2 2^n\right) \leq \int_0^{+\infty} \exp\left(-\frac{1}{4}(B\Delta)^2 2^t\right) dt
$$
$$
= \int_{(B\Delta)^2/4}^{+\infty} \frac{\exp(-u)}{\log(2)u} du
$$
$$
= \int_{(B\Delta)^2/4}^1 \frac{\exp(-u)}{\log(2)u} du + \int_1^{+\infty} \frac{\exp(-u)}{\log(2)u} du
$$
$$
\leq \int_{\min\{1,(B\Delta)^2/4\}}^1 \frac{1}{\log(2)u} du + \int_1^{+\infty} \frac{\exp(-u)}{\log(2)u} du
$$
$$
\leq -\frac{2\log(\min\{1, B\Delta/2\})}{\log(2)} + \frac{1}{4\log(2)}.
$$

$\square$

**Lemma F.4.** *Lemma 8 in [13] Let $\alpha \in (0,1)$, $c > 0$ and $d \in (0,1]$, we have*

$$
\max_{x \geq 0} \{cx^\alpha - dx\} = c^{\frac{1}{1-\alpha}} d^{\frac{\alpha}{\alpha-1}} \left(\alpha^{\frac{\alpha}{1-\alpha}} - \alpha^{\frac{1}{1-\alpha}}\right).
$$

