# OpenReview forum: "On Weak Regret Analysis for Dueling Bandits"
_NeurIPS.cc/2024/Conference — NeurIPS 2024 poster_

### Official Review · Reviewer_fTgP · 2024-06-21

**Soundness:** 3
**Presentation:** 2
**Contribution:** 3
**Rating:** 7
**Confidence:** 4

**Summary:**

The paper presents an analysis of weak regret in the context of dueling bandits, addressing the challenges posed by the non-linearity of weak loss. The authors introduce two algorithms: WR-TINF, which employs a reduction scheme to multi-armed bandits (MAB) and improves upon state-of-the-art methods, and WR-EXP3-IX, which exhibits varying performance in different instances. Additionally, the paper provides a lower bound on the weak regret that aligns with the regret bounds of the WR-TINF algorithm.

**Strengths:**

* **Originality**: The paper combines known methods like a reduction scheme to MAB [3] and decoupled exploration and exploitation in MAB [4] in a novel way to solve the largely unexplored weak regret problem. The use of the MAB Exp3-IX [5] in this setting is also novel.
* **Quality**: The theoretical analysis is rigorous, with clear proofs supporting the claims about the regret upper and lower bounds.
* **Clarity**: The paper contains intuitive discussions explaining the motivation behind both algorithms and the different instances they are suited to. Experimental evaluation helps validate these principles.
* **Significance**: This work provides interesting results for dueling bandits in the weak regret setting, which is much less explored compared to its strong regret counterpart. Specifically, it improves upon previous results in this setting [1,2] to become the new state-of-the-art, and provides the first time-independent lower bound for dueling bandits. For these reasons, and despite the weaknesses described below, I believe accepting it will contribute to the community.

[1] Chen, Bangrui, and Peter I. Frazier. "Dueling bandits with weak regret." International Conference on Machine Learning. PMLR, 2017.\
[2] Peköz, Erol, Sheldon M. Ross, and Zhengyu Zhang. "Dueling bandit problems." Probability in the Engineering and Informational Sciences 36.2 (2022): 264-275.\
[3] Saha, Aadirupa, and Pierre Gaillard. "Versatile dueling bandits: Best-of-both world analyses for learning from relative preferences." International Conference on Machine Learning. PMLR, 2022.\
[4] Rouyer, Chloé, and Yevgeny Seldin. "Tsallis-inf for decoupled exploration and exploitation in multi-armed bandits." Conference on Learning Theory. PMLR, 2020.\
[5] Neu, Gergely. "Explore no more: Improved high-probability regret bounds for non-stochastic bandits." Advances in Neural Information Processing Systems 28 (2015).

**Weaknesses:**

**Minor Issues**:
* line 53: 'in both selections'-> 'twice' will sound better.
* line 55: remove 'that'.
* line 58: 'tan'->'than'.
* line 85: the sentence that begins here is too long.
* line 89: remove 'by'.
* line 109: the reference in this line predates the ones in the previous paragraph, so 'then' should not be used.
* line 118: According to section 3.1.8 in [1], the improved regret for WS-S uses $\Delta_{k*,i}$ instead of $\Delta_{i,j}$.
* line 123: the upper bound of which algorithm is not specified. If this is WS-S, I think the power in the denominator should be 6 instead of 5 (section 3.1.8 in [1]).
* line 134: it seems  $\Delta_{k*,i}$ can be used here instead of  $\Delta_{k*,i}^2$.
* line 152: I did not find an SST assumption within the paper referenced in this line.
* line 164: perhaps it should be $\Delta_{j*(t),i}$ instead of $\Delta_{i,j*(t)}$.
* line 183: add 'and'.
* line 197: 'bandits'->'bandit'.
* line 201: $X_t$ is not defined yet.
* line 254: 'initial' should not be used here.
* line 268: $X_s(i,J_s)$ instead of $X_s(k,J_s)$.
* line 277: 'have'->'has'.
* line 278: 'ever'->'never'. 'last process' not explained.
* line 301: 'su'->'sub'. Also, as far as I understand, matrices in the family defined have a CW and satisfy the general identifiability assumption, but the reverse is not necessarily true, so this sentence is slightly confusing (perhaps the same holds with SST).
* line 309: space after '.'.
* lines 328-329: it is not clear what 'robust and conservative' means.
* line 346: 'WS-W and WR-EXP3-IX algorithms'->'algorithms WS-W and WR-EXP3-IX'.
* Many more corrections should be done in the Appendix (for example line 537 goes out of margin).

**Other Issues**:
* As rightfully acknowledged by the authors, the major issue that limits the scope of this paper is the fact that the lower bound does not fully describe the complexity of the weak-regret setting. The lower bound is proved for a narrow sub-family of instances, so while it does hold for the whole family of instances with a CW, there are other sub-families that might have a different lower bound. This is exemplified through the two proposed algorithms: while WR-TINF matches the lower bound, WR-EXP3-IX which does not match it may be better for sub-families that are different than the ones used to prove the lower bound. The fundamental issue here is that a lower bound should depend on all problem parameters.
* Similarly, since each one of the proposed algorithms performs better for different sub-families of instances, some prior knowledge about the environment is needed to choose which one is better. In other words, it seems none of the algorithms is optimal for the class of dueling bandits with a CW.
* I think that the Introduction, Contributions, and Related Works sections are too long with some repetitions that can be removed - the paper only begins introducing the main methods on page 5.
* The analysis of WR-TINF is similar to analysis methods used in previous works [2,3], but as I mentioned above I think the combination is novel.


[1] Bengs, Viktor, et al. "Preference-based online learning with dueling bandits: A survey." Journal of Machine Learning Research 22.7 (2021): 1-108.\
[2] Saha, Aadirupa, and Pierre Gaillard. "Versatile dueling bandits: Best-of-both world analyses for learning from relative preferences." International Conference on Machine Learning. PMLR, 2022.\
[3] Rouyer, Chloé, and Yevgeny Seldin. "Tsallis-inf for decoupled exploration and exploitation in multi-armed bandits." Conference on Learning Theory. PMLR, 2020.

**Questions:**

* line 122: Why is SST mentioned here instead of the less strict assumption of a total order of arms? If this refers to WS-S, the improved regret bound should hold for the latter (section 3.1.8 in [1]).
* WR-TINF uses a partial decoupling technique between exploration and exploitation, unlike the full decoupling used in the MAB case [2]. Can you explain why this is necessary? It would be simpler to just adopt full decoupling and not introduce the 'fake' draws $I_t^{'},J_t^{'}$. Is there some intuition behind this or is this the only way the reduction schemes work?
* As far as I understand, in the doubling scheme for WR-EXP3-IX, learning starts from scratch at each stage. I suppose this is due to independence issues like in standard doubling schemes, but it may harm performance. Did you try running experiments without forgetting previous stages?
* It's not clear to me in what way this algorithm mimics a random walk (line 275), could you explain this issue? Also, in line 277 could it be that the CW has a zero drift instead of a positive one? (as no other arm wins against it).
* In Figure 1a, why does WS-S perform so well despite the fact that its weak regret bounds (even for the SST case) are worse than those of WR-TINF?
* In Figures 1b,d why are the quantiles for WS-S so high?

[1] Bengs, Viktor, et al. "Preference-based online learning with dueling bandits: A survey." Journal of Machine Learning Research 22.7 (2021): 1-108.\
[2] Rouyer, Chloé, and Yevgeny Seldin. "Tsallis-inf for decoupled exploration and exploitation in multi-armed bandits." Conference on Learning Theory. PMLR, 2020.

**Limitations:**

There is no potential negative societal impact in this work. Limitations were discussed in the weaknesses section.

---

> ### Author Rebuttal · Authors · 2024-08-05
>
> We thank the reviewer for the valuable feedback.
>
> We thank the reviewer for pointing out minor issues and typos, we will correct them accordingly.
>
> ## Questions:
>
> 1. Indeed, the SST assumption is unnecessarily strong on Line 122, the total order assumption is sufficient.
> 2. **On using a partial decoupling instead of the full decoupling in WR-TINF:** We acknowledge that our sampling method may occasionally result in selecting the same arm twice ($I_t = J_t$), which is not ideal for weak regret minimization. However, WR-TINF's design ensures that the probability of this event is small enough to maintain the presented guarantees, which are optimal in scenarios that we describe. While we could modify the algorithm to prevent entirely that $I_t = J_t$, such a modification would not enhance our theoretical guarantees significantly. The introduction of the internal sampling step $(I'_t, J'_t)$ simplifies the technical analysis by ensuring that the played arms $(I_t, J_t)$ are conditionally independent given $(I'_t, J'_t)$. This conditional independence allows for a reduction to the standard MAB problem in our weak regret setting, stated in Lemma A.1. We propose including this discussion as a remark in the relevant section of our paper.
> 3. **On using data from past stages:** Indeed, at each stage, we start learning from scratch. This resulting independence leads to a cleaner technical analysis of the guarantees. While using samples from past stages does not change the theoretical results (except for an impact on multiplicative constants in the upper bound), it does have a practical impact, and we actually advocate for this approach in practice. This was reflected in some experiments conducted with WR-EXP3-IX using past data.
> 4. **On why $S(i)$ mimics a random walk:** In WR-EXP3-IX, when we fix the first action $i$ and play the second action $k$ according to the EXP3-IX algorithm, after a few rounds, the choices of $k$ concentrate on the best opponent $j^*(i)$ (corresponding to the best arm in this EXP3-IX instance). Consequently, the algorithm selects the pair $(i, j^*(i))$ in most rounds. This implies that the increments of $S(i)$ (the cumulative loss) are independent and identically distributed as $X(i, j^*(i)) - 1/2$. Therefore, $S(i)$ behaves similarly to a random walk with drift $\mathbb{E}[X(i, j^*(i)) - 1/2] = \Delta_{i, j^*(i)}$, though it is not exactly a random walk.
> 5. **On the zero gaps with the CW:** Recall that the case where all the gaps with the CW are $0$, the problem of weak and strong regret minimization is irrelevant, as the losses incurred in each round are a function of those gaps (this will lead to a $0$ loss for any played arms). On the other hand, if one of the gaps between the CW $i^*$ and an arm $i \neq i^*$ is $0$, then the guarantees of WR-EXP3-IX become loose due to the dependence on $\Delta_* = \min_{i \neq i^*} \Delta_{i^*, i}$ in the logarithmic factor in the upper bound. Note that in Assumption 2.1, we suppose that the CW has a probability of winning strictly larger than $1/2$.
> 6. **On numerical simulations:**
>
> **Variance of WS-W:** WS-W is a round-based procedure where the selected arms, "winner and challenger," duel in batches of iterations. The length of each batch increases with the number of duels won by the selected arms so far. When an arm loses, it is replaced by a contender chosen from the remaining arms. Once the set of candidate arms is exhausted, the process is repeated. In numerical experiments, particularly with a large number of arms (Scenario 3 in the simulations section), we observe that in some unfortunate cases, especially in the early stages, the CW may lose its duels. This results in a large number of iterations before it is picked again as a contender, leading to very high weak regret for the procedure. Although such outcomes are infrequent, they significantly impact the empirical variance of the weak regret of WS-W.
>
> **Strong performance of WS-W when the number of arms is small:** The guarantees on WS-W from [1] show that their upper bound has the advantage of a smaller numerical factor, whereas ours have larger numerical constants. However, this effect is negligible for large-size problems, as Figure 1(d) demonstrates that both WR-TINF and WR-EXP3-IX perform better than WS-W.
>
> [1] Bangrui Chen and Peter I Frazier. Dueling bandits with weak regret. In International Conference on Machine Learning, pages 731–739. PMLR, 2017.

---

> > ### Comment · Reviewer_fTgP · 2024-08-08
> >
> > Thank you for the well-written rebuttal. \
> > Regarding points 2 and 3, I understand that it stems from technical reasons and this is acceptable.
> > The empirical phenomena regarding WS-S are also explained well. \
> > As for point 5, I failed to notice that the WR-EXP3-IX algorithm never samples identical arms, which explains the positive drift for the CW as the first arm. \
> > While this work does not characterize the problem completely concerning the gaps, this will benefit the community as it sheds light on the open problem of weak regret optimality.\
> > I will raise my score to 7.

---

### Official Review · Reviewer_xTXf · 2024-07-02

**Soundness:** 4
**Presentation:** 4
**Contribution:** 3
**Rating:** 7
**Confidence:** 3

**Summary:**

The authors introduce two new algorithms for weak regret minimization in the setting of $K$-armed dueling bandits in order to demonstrate how the optimal strategy changes depending on how the victory probabilities of the Condorcet winner compare to the victory probabilities of the arm most likely to beat each non-Condorcet winner. The first algorithm, WR-TINF, applies online mirror descent with the Tsallis regularizer but chooses one of the arms in each duel in a way that achieves exploration while reserving the other arm for exploitation. The second algorithm, WR-EXP3-IX, applies the classic EXP3-IX algorithm repeatedly in stages where each stage focuses on a single arm with the goal of avoiding the stopping condition when the focus is on the Condorcet winner and hence being trapped in that stage where the Condorcet winner is played on every round. A lower bound on the worst-case weak regret which applies under a specific class of dueling bandit problems is presented to show that WR-TINF indeed matches that lower bound in terms of the dependence on $K$ and on the gap for the Condorcet winner (assuming that gap is constant across all arms it can duel). Numerical experiments are conducted to demonstrate the superiority of either WR-TINF or WR-EXP3-IX depending on the setup for the gaps.

**Strengths:**

# Originality
The authors primarily make use of techniques that have been applied to classic or dueling MAB problems in the past, but apply them to the under-explored problem of weak regret minimization which requires modifications to the approach and analysis. This makes the paper a sufficiently novel contribution to the dueling bandit literature. The authors differentiate their work from existing works primarily in how they exploit the entries of the gap matrix. The authors cite papers which have applied the Tsallis-inf and EXP3-IX algorithms to the classic MAB problem or the dueling bandit problem under strong regret, as well as the papers which make up the state of the art for weak regret minimization. In this way, they do a good job of citing relevant related works.
# Quality
It is evident that significant effort was made by the authors to show their work in the proofs. Furthermore, the authors are careful to explain the rationale behind their algorithmic design choices. In my judgment, this has led to a technically sound paper. As for the transparency with respect to the limitations of the work, the authors are honest about the fact that neither of the algorithms introduced will perform optimally in all cases, and that further work is required to find an algorithm which exhibits the benefits of both WR-TINF and WR-EXP3-IX at once. I consider this reasonable since the regret bound for WR-TINF already serves as a major theoretical contribution.
# Clarity
I found the authors’ explanations easy to understand aside from one point which I mention in the questions section. I did not see any mathematical errors in the main content of the paper, and I saw only a few typos in the writing. Overall, very well presented.
# Significance
The authors argue that understanding weak regret is valuable because in many practical scenarios where recommendation systems are used, the satisfaction of the user depends only on the option they choose, which we expect to be the option they prefer the most. Although uncertainty in preferences and the user’s own desire to explore can violate the assumption that the user will always choose the preferred option out of a selection, the notion of weak regret is generally closer to reality than that of strong regret, which consistently penalizes the recommendation system for options which the user does not choose. Hence, algorithms for weak regret minimization are worthy of exploration. Furthermore, the lower bound for weak regret which establishes the order-optimality of WR-TINF for a certain class of dueling bandit problems will be useful for researchers doing work in this area to judge what directions for future improvement are viable.

**Weaknesses:**

# Originality
I see no issues in terms of the originality of the work.
# Quality
I see no quality issues in the theoretical analysis of the algorithms. However, I would have liked to have seen the Modified Beat-the-Winner algorithm from [12] featured as a benchmark in the experiments, since the experimental results in [12] suggest that it outperforms WS-W.
# Clarity
Some of the steps in the proof of Lemma D.2 are hard to follow; in particular, for equation 34 it would help to see a few of the intermediate steps after applying the definition of $N_m$.

I will also point out the following typos in the paper:
- line 58 – “tan” should be “than”
- line 287 – “Comparaison” should be “Comparison”
- line 301 – “su-optimal” should be “sub-optimal”
- line 308 - should be “linear scaling with K is optimal in this case” (you are missing the "is")
- line 557 – “having” should be “have”
- line 668 - “we the proof” should be “with the proof”
# Significance
Because this paper is primarily applying known techniques to a problem setting where they have not been fully exploited yet, its significance is somewhat limited to that problem setting. However, this is a perfectly valid approach in cases where the state-of-the-art for that problem setting can be advanced significantly.

**Questions:**

In the explanation of algorithm 2 at the start of page 8, I don’t see why $\Delta_{sub}$ wouldn’t always be greater than or equal to $\Delta_{cw}$. If $\Delta_{sub}$ is the maximum gap that any arm can have over arm $i$ and $\Delta_{cw}$ is the gap for a specific arm, namely the Condorcet winner, how would it be possible for their ratio to be smaller than 1? The example given in lines 161 to 169 makes more sense to me because $\Delta_{sub}$ is defined such that it clearly does not depend on $\Delta_{cw}$. This seems like a miscommunication rather than a technical oversight; maybe you meant for $\Delta_{sub}$ to be the max over all arms excluding $k^*$?

**Limitations:**

Since this paper presents foundational research, there are no societal impacts to consider. The main limitation with the algorithms that the authors present is addressed in Section 7 of the paper, as mentioned in the strengths section.

---

> ### Author Rebuttal · Authors · 2024-08-05
>
> We thank the reviewer for the valuable feedback.
>
> ## Weaknesses:
>
> **Quality:** We performed additional experiments using the MBTW algorithm (from [1] below) under the same scenarios outlined in our Numerical Simulations section. The simulation results are presented in Figure 1 of the global rebuttal.
>
> Firstly, it is important to note that there are no theoretical guarantees developed for the MBTW algorithm. Nevertheless, we included MBTW in our experiments to provide a comprehensive comparison.
>
> The experimental results, detailed in Figure 1 of our global rebuttal, indicate that while the MBTW algorithm performs similarly to WR-EXP3-IX and WR-TINF in scenarios with a moderate and large number of arms (Scenarios 3 and 4), its performance is unstable in Scenario 1. Specifically, we observed high variance in results and instances where the regret diverged significantly, resulting in very large regret values in some cases. This particular experiment was conducted multiple times, yielding consistent outcomes.
>
> **Clarity:**
> Thank you for your suggestion and for pointing out the typos. We will correct them. Additionally, we will add intermediate steps to the proof of Lemma D.2 to make it clearer and easier to follow.
>
> ## Question:
>
> You are correct regarding this point. In the discussion of Algorithm 2 (beginning of page 8), $\Delta_{\text{sub}}$ should be defined as the maximal gap between sub-optimal arms (as mentioned in the paragraph by words): $\Delta_{sub} = \max_{i,j \neq k*} \Delta_{i,j}$ instead of $\Delta_{sub} = \max_{j} \Delta_{i,j}$. We apologize for the typo and will correct it.
>
> [1] Erol Peköz, Sheldon M Ross, and Zhengyu Zhang. Dueling bandit problems. Probability in the Engineering and Informational Sciences, 36(2):264–275, 2022.

---

> > ### Comment · Reviewer_xTXf · 2024-08-09
> >
> > I thank the authors for their rebuttal, and I appreciate them taking the time to include the MBTW algorithm in their experiments. Overall, I believe my original score remains appropriate.

---

### Official Review · Reviewer_bFCz · 2024-07-11

**Soundness:** 3
**Presentation:** 4
**Contribution:** 3
**Rating:** 7
**Confidence:** 3

**Summary:**

This paper addresses weak regret minimization. The authors demonstrate a lower bound result in terms of gaps between the Condorcet winner and the sub-optimal arms. Furthermore, they propose the WR-TINF algorithm, which achieves this optimal regret when the optimality gap is sufficiently large. Additionally, they introduce the WR-EXP3-IX algorithm, which outperforms WR-TINF when the optimality gap is negligible.

**Strengths:**

1. The authors proposed two algorithms focusing on minimizing weak regret

2. The authors provided both upper and lower bounds rigorously.

3. Both algorithms have special advantages. For instnce, the former algorithm is optimal in some instances. The latter algorithm is more powerful when the optimality gap is negligible.

**Weaknesses:**

1. Too long and comlicated structure of proofs

2. some typos and unclear notations (with equations out of range)

**Questions:**

1. Lemma B.4 in Line 545 used the learning rate $\eta_t=\frac{2\beta}{\sqrt{t}}$ but the equation (8) in Algorithm 1 used $\frac{1}{\eta_t}$. Is  this  corrct?

2. tan in line 58 should be than. Is this right?

3. In Algorithm 1 WR-TINF, else statement seems to be unnecessary because $I_t$ and $J_t$ are already selected.

4. If $J_t$ is sampled by $r_t$ in Algorithm 1, and still $I_t$ equals to $J_t$, It is curious whether $J_t$ is sampled again or not.

5. If the Condorcet winner is unique and there is a large optimality gap, I suspect that most of the second arms sampled by $r_t$ will still be the Condorcet winner after a sufficient number of rounds. If so, the WR-TINF algorithm may become sublinear in terms of strong regret (i.e., its exploration ability may eventually disappear). Please seek the authors' opinion on this matter. i.e., I am curious whether WR-TINF is still effective to the strong regret.

6. The equation in Line 537 is outside the scope of the paper.

7. the mearning of $\lceil * \rceil$ is unclear in Line 507.

**Limitations:**

The authors clearly specified their paper's limitation (and solving the limitation will be difficult).

---

> ### Author Rebuttal · Authors · 2024-08-05
>
> We thank the reviewer for the valuable feedback.
>
> ## Weaknesses:
>
> 1. We apologize for the typos and the out-of-range equations. We will correct the typos we have identified as well as those noted in the reviews.
>
> ## Questions:
>
> 1. **On the correctness of Equation 8:** Both the statement of Lemma B.4 and equation (8) in Algorithm 1 are correct. We employed a version of online mirror descent with Tsallis regularization as analyzed in [1] (specifically, refer to Section 3.2 in [1]).
> 2. Thank you for spotting this typo, we will correct it.
> 3. **On resampling $(I_t,J_t)$ in the else statement of WR-TINF:** In WR-TINF, we initially sample two arms $(I'_t, J'_t)$ without playing them. Based on the outcome (whether $I'_t = J'_t$ or not), we then specify the distribution for the arms to be played $(I_t, J_t)$. The reason for resampling $I_t$ and $J_t$ from the same distribution (the else statement) is to enhance clarity by distinguishing between the internal sampling step of $(I'_t, J'_t)$ and the step where we sample the arms $(I_t,J_t)$ that are actually played. Additionally, this is simpler technically, as the played arms $I_t$ and $J_t$ remain independent given $I'_t$ and $J'_t$ - see Algorithm 1.
> 4. **On the possibility of having $I_t=J_t$:** You are correct in noticing that with our sampling scheme, there is still a possibility of playing the same arm twice ($I_t = J_t$), which is not ideal for minimizing the weak regret as discussed. However, the rationale behind WR-TINF is that, contrary to algorithms designed for minimizing the strong regret, our scheme makes the probability of the event $I_t = J_t$ low enough to ensure the presented regret upper bound guarantees - which are optimal in the cases that we describe. We could modify the algorithm to ensure that this event never occurs, for example by repeatedly sampling $(I_t, J_t)$ until they are different. However, this would not improve significantly the theoretical guarantees and would complicate the analysis of the algorithm.
> 5. **On the performance of WR-TINF in the strong regret setting:** We agree that if the optimality gaps of the Condorcet winner are large, then the sampling distributions $p_t$ and $r_t$ will quickly concentrate on the optimal arm, leading to small losses for the strong regret in this specific case. Note that $r_t$ will concentrate slower than $p_t$ on the Condorcet winner. It is possible that in this scenario, one can derive sublinear regret for the strong regret. However, we believe that it will not achieve the optimal guarantees for the strong regret, which are logarithmic in $T$.
> 6. Thank you for spotting this formatting issue, we will correct it.
> 7. We apologize for the typo, instead of $\bar{T} := \lceil*\rceil\frac{256K}{\Delta_*^2}$, we meant $\bar{T} := \lceil\frac{256K}{\Delta_*^2}\rceil$.
>
> [1] Julian Zimmert and Yevgeny Seldin. Tsallis-inf: An optimal algorithm for stochastic and adversarial bandits. The Journal of Machine Learning Research, 22(1):1310–1358, 2021.

---

> ### Comment · Reviewer_bFCz · 2024-08-12
>
> Thank you for your explanation.
>
> Aside from such minor issues (e.g., typos), I believe the paper is excellent in terms of motivation (weak regret) and is mathematically robust. After reviewing the appendix again, I found no issues with the overall flow. The additional experiments further strengthened my confidence in the paper.  I will raise my score to 7.

---

### Author Rebuttal · Authors · 2024-08-05

We thank the reviewers for their important feedback. As suggested by reviewer xTXf, we conducted experiments including the Modified Beat The Winner (MBTW) algorithm from [1]. While its performance is good in some scenarios (comparable to WR-EXP3-IX and WR-TINF), it suffers from instability in other scenarios, as shown in Figure 1 of the attached file.

[1] Erol Peköz, Sheldon M Ross, and Zhengyu Zhang. Dueling bandit problems. Probability in the Engineering and Informational Sciences, 36(2):264–275, 2022.

---

### Decision · Program_Chairs · 2024-09-25

**Decision:**

Accept (poster)

**Comment:**

All referees unanimously agree that the paper makes valuable contributions in terms of providing insight about fundamental limits of (weak) regret performance in dueling bandits with only a (weak) Condorcet winner, exposing the influence of the preference matrix on the regret, and providing new algorithms for dueling bandits. In view of this, the paper is recommended for acceptance.